# SheetCopilot: Bringing Software Productivity to the Next Level through Large Language Models

Hongxin Li[*1,2], Jingran Su[*3,4], Yuntao Chen[†3], Qing Li[†4], and Zhaoxiang Zhang[†1,2,3,5]

[1]School of Artificial Intelligence, University of Chinese Academy of Sciences (UCAS)
[2]State Key Laboratory of Multimodal Artificial Intelligence Systems (MAIS), Institute of Automation, Chinese Academy of Sciences
[3]Center for Artificial Intelligence and Robotics, HKISI, CAS
[4]The Hong Kong Polytechnic University
[5]Shanghai Artificial Intelligence Laboratory

## Abstract

Computer end users have spent billions of hours completing daily tasks like tabular data processing and project timeline scheduling. Most of these tasks are repetitive and error-prone, yet most end users lack the skill to automate these burdensome works. With the advent of large language models (LLMs), directing software with natural language user requests become a reachable goal. In this work, we propose a SheetCopilot agent that takes natural language task and control spreadsheet to fulfill the requirements. We propose a set of atomic actions as an abstraction of spreadsheet software functionalities. We further design a state machine-based task planning framework for LLMs to robustly interact with spreadsheets. We curate a representative dataset containing 221 spreadsheet control tasks and establish a fully automated evaluation pipeline for rigorously benchmarking the ability of LLMs in software control tasks. Our SheetCopilot correctly completes 44.3% of tasks for a single generation, outperforming the strong code generation baseline by a wide margin. Our project page: https://sheetcopilot.github.io/.

## 1 Introduction

The ability to intuitively direct sophisticated software through natural language has long been an aspiration pursued across generations. With the emergence of Large Language Models (LLMs) that can comprehend and respond to human directives, this vision is within closer reach now than ever. LLMs augmented with tools [24, 25, 20, 26, 2, 27] and reasoning abilities [32, 18, 30, 31] have shown promising results in recent research.

While LLMs continue advancing rapidly, their ability to interoperate seamlessly with existing software tools remains under-explored and not fully understood. Enabling LLMs to harness the rich functionality of countless existing software tools could unlock nearly limitless potential [21].

Progress in endowing LLMs with the ability to direct complex software systems has been hindered by the lack of both a standardized framework for model-application interaction and a comprehensive benchmark for evaluating their performance. Several challenges exist in designing a framework to facilitate interaction between LLMs and sophisticated software applications, including 1) Translating the complex internal state and vast functionality of applications into text forms comprehensible for

---

[*] Equal contribution. E-mails: lihongxin2021@ia.ac.cn, jing-ran.su@connect.polyu.hk
[†] Equally advising corresponding authors. E-mails: zhaoxiang.zhang@ia.ac.cn, csqli@comp.polyu.edu.hk, chenyuntao08@gmail.com

37th Conference on Neural Information Processing Systems (NeurIPS 2023).

models [32]. This requires determining how to systematically represent software interfaces and logic through natural language; 2) Enabling models to generate software commands and parameters accurately and safely [9, 2]. Mechanisms must exist for validating, debugging, and as needed, rejecting or revising model outputs to avoid undesirable operations or states; 3) Providing models with means of monitoring software state changes, exceptions, and errors during multi-step tasks so that these models can respond appropriately. Models are required to understand software feedback and diagnose issues, and adjust directives as needed to robustly accomplish goals. In addition, enabling LLMs to direct complex software also requires curating datasets that capture the diversity and ambiguity of real-world language use, as well as developing automated techniques to reliably evaluate model performance at scale [7].

To systematically investigate the substantial challenges in developing natural language interfaces for software control, a robust application platform is required; as the most pervasive and multi-functional productivity tool, the spreadsheet serves as an ideal substrate for this work. To this end, we propose a general framework for facilitating interaction between language models (LMs) and software applications, along with an agent called **SheetCopilot**. As shown in Fig. 1 SheetCopilot understands high-level spreadsheet manipulation requests expressed in natural language. It decomposes these complex requests into step-by-step plans, and issues commands to automatically carry out the necessary operations using the spreadsheet application. In addition to our spreadsheet-manipulating agent, SheetCopilot, we propose a dataset consisting of complex, interactive spreadsheet manipulation requests expressed through natural language and an evaluation framework with automated metrics to assess how accurately models comprehend requests, devise optimal plans, and perform operations through the spreadsheet interface. We believe robust measurement is key to accelerating progress in this area.

Our agent SheetCopilot achieved substantial capabilities for guiding spreadsheet software through natural language. It generated fully executable command sequences for 87.3% of problems in our benchmark suite and produced completely correct solutions for over 44.3% of tasks—surpassing the traditional programming approaches by a wide margin. To rigorously assess model performance, we curated a dataset of 221 representative spreadsheet tasks collected from `superuser.com`, including verified solutions created by the authors for each task.

We present three primary contributions to the goal of achieving sophisticated interfaces between language models and traditional software applications:

- We proposed a general framework for facilitating model-software interaction along with SheetCopilot, an agent specialized for spreadsheets that translates high-level, task-oriented requests expressed in natural language into executable command sequences.

- We developed comprehensive resources for systematically evaluating model and interface performance, including a benchmark suite of interactive spreadsheet tasks reflecting real-world requests and a fully automated pipeline for measuring how accurately models comprehend complex prompts, devise optimal plans and execute operations through the software interface.

- We conducted an in-depth assessment benchmarking the abilities of leading LLMs in this challenging domain. The experiments show that LLMs equipped with our method significantly outperform the strong code generation baseline.

## 2 Related Works

**Tool-augmented Large Language Models** Recently, the impressive performance of LLMs has sparked significant interest in exploring the task-planning capabilities of LLMs in various fields. Benefitting from the internalized knowledge about the world [1], a number of works have managed to enable LLMs to solve tasks by following instructions. One line of research [14, 3, 16, 15, 8] has utilized prompt engineering to elicit a sequence of mid-level steps from LLMs for household robotic tasks. To ground LLMs in the real world, these works have used auxiliary models [3, 16, 15] or trained LLMs via mixing visual-language data and embodied data [8]. Another promising direction is to connect LLMs with external tools [25], such as a web browser [22], HuggingFace model hub [26], chemical software [2], PowerPoint [20], and even a tool library [25, 24]. These works employ LLMs to generate action sequences which are further parsed into API calls of the tools. Compared with

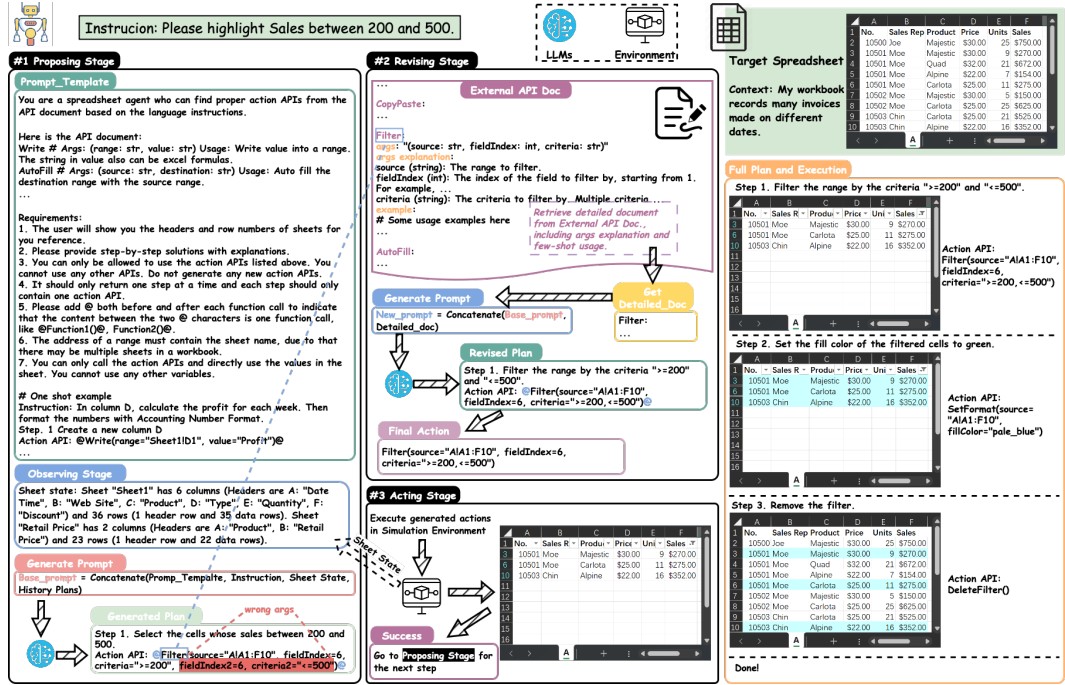

Figure 1: We maneuver SheetCopilot to control software such as Microsoft Excel, generate step-by-step solutions fulfilling the user's requirements. In each step, SheetCopilot plans an initial atomic action according to the sheet state and then revises this step using the external document which provides the action usage and examples. Finally, the action with its arguments is extracted from the revised step and submitted to the simulation environment for execution. The entire process on the right shows that SheetCopilot successfully solves the task specified in the instruction using the provided available atomic actions.

these works, our work is targeted at spreadsheet manipulation, which is a common demand in almost all scenarios (e.g. economics, management, and research). To the best of our knowledge, limited research has been conducted on benchmarking the capability of LLMs for spreadsheet manipulation.

**Natural Language Processing (NLP) for Spreadsheets** Several studies [12, 11, 28, 6, 17] have investigated the feasibility of using NLP methods to guide the manipulation of Excel sheets. An early work is Flash Fill [11], which automates string processing tasks using program synthesis by example. NLyze [12] utilizes a translation algorithm to convert a user's natural language instruction to a ranked set of likely programs. Inspired by the success of Codex [5] and AlphaCode [19], one recent study [17] has explored the use of LLMs for generating Excel formulas given natural language descriptions of the desired output. They compared the performance of several state-of-the-art LLMs, including GPT-3 and T5, and found that these models can generate accurate formulas with high efficiency. However, this study focused on formula generation rather than general sheet control tasks. In this paper, we aim to address this gap by benchmarking the capability of LLMs for sheet control tasks.

## 3 Dataset and Evaluation

Prior research on language interfaces for spreadsheet control [12, 6, 17] has focused primarily on limited subsets of tasks like formula generation and lacked comprehensive, standardized means of evaluation. To address this issue, we aim to construct a high-quality evaluation benchmark as a foundation for assessing the spreadsheet control capabilities of LLM-based agents.

Our dataset compilation procedure incorporates gathering tasks and worksheets from the Internet, filtering low-quality or irrelevant tasks, consolidating redundant entries, adapting seed tasks, and manually annotating a core set. The end product is a comprehensive and cleanly-labeled collection of spreadsheet-related tasks. We also report statistics and analysis to characterize the dataset properties,

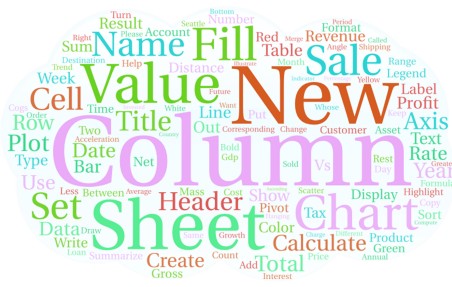 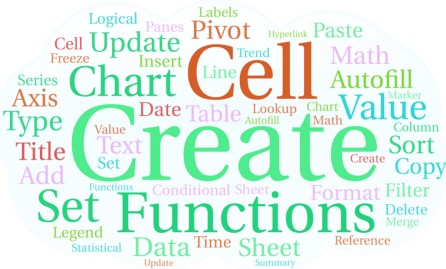

| Wordcloud of the instructions | Wordcloud of the atomic actions |

Figure 2: Dataset overview. We present an overview of the core set by showing the wordclouds of the instructions and involved atomic actions. The two clouds show that the core set contains diverse tasks that involve various spreadsheet operations.

guide future work, and set initial baselines. Moreover, we develop an automated, reproducible evaluation framework closely tailored to our curated natural language spreadsheet control dataset. This enables systematically assessing model abilities, gaining insights into current limitations, and driving continued progress in this domain.

## 3.1 Diverse Seed Task and Workbench Collection

We first scrape all questions with spreadsheet-related tags on `www.superuser.com` and obtain a raw dataset comprising ∼16k question and answer (Q&A) pairs. Sourcing questions from SuperUser ensures our task dataset is both comprehensive and representative. As not every question represents a sheet manipulation task we apply keyword-based and LLM-based filters to remove Q&A pairs unrelated to spreadsheet automation, resulting in a remain of ∼13k pairs. To analyze the distribution of the dataset, we define six task categories: Entry and Manipulation, Management, Formatting, Charts, Pivot Tables, and Formulas. We label exemplar Q&A pairs with at least one category to prompt the language model to categorize each pair, as pairs may belong to multiple categories. To identify representative Q&A pairs, we embed and cluster pairs within each unique category combination. We then choose 67 pairs representing the clustering centers and involving operations supported by our evaluation environment. The spreadsheet tasks described in these pairs are regarded as the seed tasks which capture the most important patterns of our dataset.

To evaluate LLMs, we also collect 28 real-world spreadsheets as our workbench by 1) downloading practical sheets from the Internet, and 2) Generating typical daily-use sheets by hand. These sheets represent common uses such as analyzing sales data, calculating financial metrics, and visualizing data with charts.

## 3.2 Core Set Collection

The seed tasks cannot be directly used since their original sheets differ from the evaluation sheets. We propose collecting a core dataset by adapting and simplifying the seed tasks to bridge this gap.

**Adaptation**. Inspired by self-instruct [29], we prompt an LLM to adapt the seed tasks according to the detailed descriptions of the evaluation sheets. Specifically, GPT-4 is prompted to change the manipulated elements in the seed tasks to generate new tasks compatible with the evaluation sheets. For instance, GPT-4 can change the data types required to be set or ranges to be modified in the original seed task. In this step, 1669 new task instructions are generated (See Tab. D for examples).

**Simplification**. The adaptations are likely to mention specific formulas and operations. To address this issue, we prompt an LLM to simplify each task by replacing specific expressions with natural spoken language so that the task instruction reads like the fashion of a non-expert user. This step reduces the average token length from $47.1$ to $33.8$[1].

---

[1]We followed the instruction on https://github.com/openai/tiktoken to count the token number using the model "gpt-3.5-turbo".

To collect a core set, the authors select several tasks for each category combination from the simplified tasks. The authors also act as non-expert users to compose more tasks to enrich the core set, obtaining 221 tasks in total. Finally, multiple reference solutions are collected as the ground truth for each task. See Fig. 2 for the dataset statistics and more in the appendix.

### 3.3 Task Evaluation by Execution

It is hard to evaluate solutions generated by LLMs through verbatim comparison, as it is likely that multiple solutions can successfully complete a task. A viable approach is assessing whether the final sheet state after executing the solution meets the task instruction. We only assess the necessary properties required for the ground truth spreadsheet's operation. For example, in the task "Plot a line chart with the X-axis showing the week and the Y-axis showing sales", we only consider properties related to the chart, ignoring other aspects. To assess an LLM-generated solution, we evaluate the consistency of the necessary properties between the spreadsheet resulting from executing this solution and the ground truth spreadsheet in our evaluation environment. If the necessary properties of the resulting spreadsheet fully match any potential ground truth candidate, the associated solution is deemed correct.

## 4 Method

SheetCopilot enables natural language interactions with spreadsheets. It takes spreadsheets and user tasks described in natural language as input and generates plans to modify spreadsheet contents and create diagrams or tables. We adopt an in-context learning framework inspired by models such as GPT-3 [4]. We propose "atomic actions" - a set of virtual APIs representing common spreadsheet functions. These actions allow language models to accurately interact with the spreadsheet software. We also propose a state machine-based task planning framework to handle the complex, multi-turn interaction between the language models and the spreadsheets. The atomic actions and state machine-based planning framework enable language models to effectively and robustly direct spreadsheet software through natural language.

### 4.1 Prompting LMs as a SheetCopilot

We design a systematic prompt template to turn LMs into copilots as shown in the left of Fig. 1. Our prompt consists of a general role description, a list of atomic actions with arguments, a set of output requirements, and a multi-round interaction example between a user and an assistant.

The general role description serves as an anchor for enabling LMs to understand the context. A list of atomic actions provides LMs with the interface information needed for task planning. The output requirement tells LMs how to generate texts that can be programmatically extracted and executed. The multi-round example hints LMs how the observe-propose-revise-act loop appears and improves the overall planning quality.

### 4.2 Atomic Action as A Bridge for LMs and Software

State-of-the-art LMs have shown the superb ability to generate detailed plans for household tasks [16], software control [20], and debugging [5]. However, the generated plans are in natural language which is easy for humans to read but not directly admissible for machine execution.

To overcome the limitation mentioned above, we propose to model the functionalities of existing spreadsheet software as a set of virtual APIs called atomic actions. An atomic action is comprised of an API name, a typed argument list, a usage document string, and several usage examples. These atomic actions can be implemented on different spreadsheet platforms. The example implementations in Tab. H of the appendix show that the atomic actions involve cell value modification, formatting, sheet management, formula and functions, charts, and pivot tables.

Choosing proper atomic action granularity is crucial, as actions must be expressive yet concise to fit in the LM context windows. We determine our atomic actions as follows: 1) Extract all actions involved in the top SuperUser spreadsheet Q&As; 2) Embed and cluster the extracted actions into candidates; 3) Select the minimum set of actions covering all the tasks we collected in Sec. 3.1.

**Relation to Agents Generating VBA Codes** LMs are also capable of generating machine-readable codes [5]. This approach is especially tempting for Microsoft Excel as it comes with an embedded script language called Visual Basic for Applications(VBA). However, the code generation approach faces challenges from both the LMs side and the spreadsheet software side. On the code LMs side, the existing training corpus [10, 13, 5] for code LMs hardly contains VBA source files as it is only a niche programming language compared with C++ or Python. On the spreadsheet software side, software such as Google Sheets, Libre Office, and WPS either do not support VBA at all (Google Sheets) or only support a limited subset of VBA functions (Libre Office and WPS). Therefore, we advocate a more software-agnostic approach that does not rely on embedded programming language support.

### 4.3 State Machine-based Task Planning

A normal spreadsheet task usually involves several steps, while a sophisticated one often requires over ten steps. Open-loop planning - directly generating a complete task plan from the instruction - becomes exponentially harder as steps increase. Each step changes the sheet state so the correct step $T + 1$ relies on perfectly understanding how the sheet state changes after the previous $T$ steps. As tasks become more complex, open-loop planning struggles.

We propose a state machine-based planner which revises the plan according to feedback from either LMs or software. Our planner is divided into observing, proposing, revising, and acting stages. The state transition between these stages will be described below. Due to the page limit, please refer to the appendix for examples of complete planning logs.

**Observing Stage** In this stage, we add a brief description of the sheet state $S_t$ to the query, providing information such as the name of each column and the total number of rows for LMs to determine atomic action arguments. This allows LMs to generate solutions in a closed-loop manner by observing the previous actions' consequences without implicitly modeling sheet states.

**Proposing Stage** In this stage, we concatenate the system prompt $P$, the initial task instruction $I$, the sheet state $S_t$ and the planning history $H_t$ and ask the LMs to plan the next atomic action $A_{t+1}$.

$$A_{t+1} = \text{Validate}(R_{t+1}) = \text{Validate}(\text{LanguageModel}(P, I, S_t, H_t)). \tag{1}$$

The direct response $R_{t+1}$ from the language model is not always convertible to an admissible atomic action $A_{t+1}$. Common errors found in the validating step include missing the format requirement, hallucinating undefined actions, and incorrectly determining action parameters.

**Revising Stage** Two ways are adopted to revise a proposed atomic action: a feedback-based one and a retrieval-based one. Feedback-based revision utilizes the error feedback from both the atomic action validation and the spreadsheet software execution. For example, if the atomic action validating step detects a hallucinated atomic action, a new prompt will be created to inform the LM of this error and to ask it to reiterate the available atomic actions. Additionally, we use retrieval-based revision to supply the LM with detailed external knowledge that does not fit in the system prompt due to the context window limit. For example, if the LM uses an atomic action with wrong arguments, a detailed document containing the argument descriptions and usage examples of this action is provided in the new prompt to enhance the probability of the LM correctly determining the atomic action arguments. This process resembles how a human programmer behaves when encountering less familiar APIs.

A special case in the revision stage is that after being supplied with more information about the initially proposed atomic action, the LM suddenly finds that it has chosen a wrong action and decides to return to the revising stage.

**Acting Stage** After the proposing and revising stages, the atomic action $A_{t+1}$ is submitted to the spreadsheet software for execution.

$$S_{t+1} = \text{SpreadSheetEnv}(A_{t+1}, S_t). \tag{2}$$

The planning history $H_t$ is updated if the execution succeeds,

$$H_{t+1} = H_t \cup \{A_{t+1}, S_{t+1}\}. \tag{3}$$

If the software reports a run-time error, the state machine will return to the proposing stage to prompt the LM to re-plan according to the error feedback.

Table 1: Performances of the compared LLMs and a VBA-based method. The three LLMs exhibit impressive Exec@1 and Pass@1, with GPT-3.5-Turbo achieving the highest Exec@1 and GPT-4 obtaining the best Pass@1 and efficiency. With our method, GPT-3.5-Turbo outperforms the method that generates and runs VBA code by a large margin.

| Data | Models | Exec@1↑ | Pass@1↑ | A50↓ | A90↓ |
|------|--------|---------|---------|------|------|
| 10% | GPT-3.5-Turbo | **85.0%** | 45.0% | 2.00 | 4.50 |
| 10% | GPT-4 | 65.0% | **55.0%** | **1.33** | **2.00** |
| 10% | Claude | 80.0% | 40.0% | 1.50 | 4.40 |
| 100% | GPT-3.5-Turbo | 87.3% | 44.3% | 1.50 | 3.00 |
| 100% | VBA | 77.8% | 16.3% | - | - |

## 4.4 Hallucination Mitigation

To enable the state machine to less frequently return to the proposing stage due to hallucination-induced errors, we adopt the following means.

**Output Formatting** The underlying functions of atomic actions require precisely formatted planning results. However, we found that LMs probably generate semantically correct yet inadmissible action plans as shown in Fig. 1. Therefore, we require LMs to wrap actions with special tokens (e.g. @) and detect the tokens in the output to check whether the output is correctly formatted.

**Atomic Action Disambiguation** The internalized knowledge in LMs is likely to be confused with the atomic action definitions in the document. Due to this conflict, LMs are prone to self-delusion, which means that it hallucinates undefined actions or adds illegal action arguments [23, 14]. To tackle this problem, the atomic action names are substituted with a set of synonyms that are far away from the official names in an embedding space. For instance, Write and SetConditionalFormat are substituted with RangeInputValue and FormatWithRules, respectively (See the details in the appendix).

## 5 Experiments

The goals of our experiments are threefold: (i) compare representative LLMs on the proposed dataset; (ii) demonstrate that the proposed method improves the success rate and efficiency over a simple baseline; (iii) show the flexibility and stability of our method.

### 5.1 Benchmark Protocol

**Dataset** The 221 tasks introduced in Sec. 3.2 are used to conduct the following experiments.

**Metrics** Exec@1 measures the proportion of solutions executed without throwing exceptions. Pass@1 is used to evaluate functional correctness [5]. A generated plan is considered correct if the final sheet state completely fulfills the task requirements. Beyond correctness, we propose A50 and A90 scores to measure solution efficiency. These divide the number of atomic actions in a generated plan by the number in the ground truth and then calculate the 50th and 90th percentiles over all tasks. Lower A50 and A90 scores indicate that the LLM tends to use fewer actions to complete a task.

**Models** We adopt leading large language models with public API access, including GPT-3.5-Turbo/GPT-4 from OpenAI and Claude v1 from Anthropic. Details of the models and hyper-arguments used for generation could be found in the appendix.

### 5.2 Comparing Task Planning Ability of Different LLMs

We compare the three LLMs on the proposed dataset with the same token limit of 4096. For less accessible LLM APIs like GPT-4 and Claude, only 10% of the dataset is used for evaluation. We have maintained the diversity of this subset to avoid data distribution shift (see the appendix for details). The results in Tab. 1 show that GPT-4 demonstrates its strong planning capability by significantly outperforming both GPT-3.5-Turbo and Claude in the Pass@1 and A50/A90. To explain why GPT-4 is inferior in Exec@1, we check the results and find that it is mainly because GPT-4 exceeds the

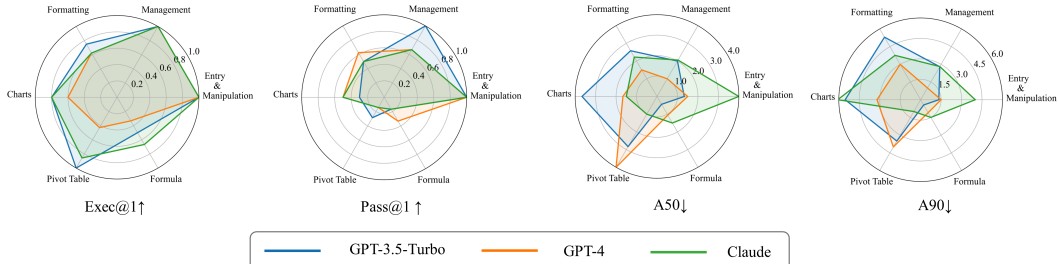

Figure 3: The four metrics decomposed in the six categories. The two GPT models both achieve 100% Exec@1 and Pass@1 in the Management and Entry & manipulation categories. The three models obtain their own best efficiency in different categories, suggesting that it is difficult for these models to excel in all task categories.

Table 2: Ablation studies of the observe-propose-revise-act framework proposed in Sec. 4.3. The sheet state and error feedback increase the Exec@1 and Pass@1 when individually applied (rows 3, 4 vs. 2) and bring a significant improvement when both are applied (row 7 vs. 2). Inserting the external atomic action document with usage examples boosts the Exec@1 and Pass@1 and increases efficiency (row 2 vs. 1 and row 7 vs. 5). The synergy of the four components witnesses a large increase (30.7%) in Exec@1 over the baseline (row 7 vs. 1).

| No. | State feedback | Error feedback | External doc. | Usage examples | Exec@1↑ | Pass@1↑ | A50↓ | A90↓ |
|-----|----------------|----------------|---------------|----------------|---------|---------|------|------|
| 1 | | | | | 56.6% | 18.6% | 1.50 | 4.00 |
| 2 | | | ✓ | ✓ | 67.9% | 18.6% | 1.00 | 2.50 |
| 3 | ✓ | | ✓ | ✓ | 75.6% | 24.4% | 1.50 | 3.00 |
| 4 | | ✓ | ✓ | ✓ | 92.8% | 23.5% | 1.00 | 2.00 |
| 5 | ✓ | ✓ | | | 68.3% | 18.1% | 1.50 | 3.78 |
| 6 | ✓ | ✓ | ✓ | | 85.5% | 28.1% | 1.50 | 3.00 |
| 7 | ✓ | ✓ | ✓ | ✓ | 87.3% | 44.3% | 1.50 | 3.00 |

token limit when solving difficult tasks although it has generated correct mid-steps. In contrast, GPT-3.5-Turbo and Claude generate short but incorrect plans for most of these difficult tasks. Claude is slightly worse than GPT-3.5-Turbo in Exec@1 and Pass@1 but exhibits better A50/A90, which shows that Claude is a strong competitor of GPT-3.5-Turbo. See more detailed failure analysis in Sec. F.

To evaluate the category-specific performances, We further break down the subset into the six categories (defined in Sec. 3.1). The four metrics in each category are illustrated in Fig. 3. The radar charts demonstrate that the two GPT models both achieve 100% Exec@1 and Pass@1 in the Management and Entry & manipulation categories. Interestingly, The three models exhibit different patterns of A50/A90: GPT-3.5-Turbo, GPT-4, and Claude reach their best efficiency in the Formula, Management, and Pivot Table category, respectively. This suggests that it is difficult for these models to excel in all task categories.

### 5.3 Ablation Studies of State Machine

We conduct ablation studies for GPT-3.5-Turbo on the full dataset to analyze the impact of the two types of feedback and external document insertion. The results are shown in Tab. 2.

**A) Closed-loop control generally boosts functional correctness** Individually adding the sheet state feedback at the proposing stage increases the Exec@1 and Pass@1 (rows 3 vs. row 2) since the model no longer needs to implicitly infer the sheet state. Individually adding error feedback obtains the highest Exec@1 (row 4 vs. row 2) as a longer context window can be used to re-plan without the sheet state feedback, increasing the probability of completely finishing a plan. The combination of the two feedback types further improves Pass@1 but at the cost of slightly increasing A50 and A90 (row 7 vs. row 2), probably because the model solves more difficult tasks but with plenty of steps. It is noticeable that without the external document and usage examples, the improvement in Exec@1 becomes narrow and Pass@1 even drops slightly (row 5 vs. row 1). This is because the

Table 3: Ablation study of the atomic action names. Utilizing the synonyms far away from the official names brings an increase in Pass@1 and slightly better efficiency (lower A50).

| Models | Exec@1↑ | Pass@1↑ | A50↓ | A90↓ |
|---|---|---|---|---|
| Official names | 87.3% | 44.3% | 1.50 | 3.00 |
| Synonyms | 86.9% | 45.2% | 1.33 | 2.79 |

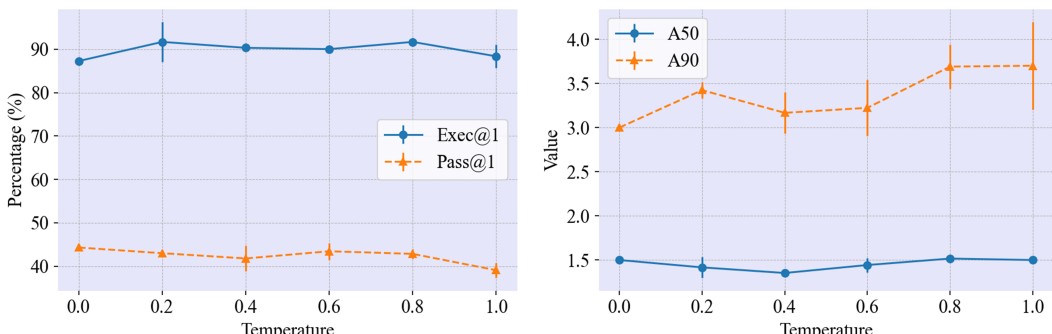

Figure 4: Stability experiments results obtained by conducting evaluation 3 times for each temperature except 0.0. The line charts show that SheetCopilot achieves stable performances even if the GPT-3.5 API temperature changes from 0.0 to 1.0.

simple action list in the initial prompt (shown in Tab. I) fails to provide sufficient information about how to determine correct action arguments even if a detailed sheet description is provided.

**B) Inserting the external document improves both functional correctness and efficiency** Merely adding the external atomic action document enjoys clear improvements of **17.2%** in Exec@1 and **10.0%** in Pass@1 (row 6 vs. row 5). This result demonstrates that presenting this document to the model enables it to less frequently hallucinate illegal actions (which improves Exec@1) and to more accurately determine the action arguments according to the sheet state (which improves Pass@1). Further adding usage examples in the document reaches the highest performance (row 7). Additionally, without any feedback, the improvements over the baseline become narrower (row 2 vs. row 1) since it is hard to determine correct action arguments without knowing the exact properties of the spreadsheet elements even if more details of atomic actions are provided. Adding the external document and usage examples reduces A50/A90 (row 2 vs. row 1 and rows 6, 7 vs. row 5), showing that more information about atomic action usage helps to avoid redundant steps.

**C) SheetCopilot surpasses the VBA-based method** To further demonstrate the advantages of our method, we compare it with a method that generates and runs VBA code. Tab. 1 shows that SheetCopilot with GPT-3.5-Turbo as its backend outperforms the VBA-based method by a large margin, increasing the Exec@1 by **9.5%** and Pass@1 by **28.0%**. This result shows that prompting powerful LLMs with our method to control spreadsheets is a better alternative to directly translating natural language requests to VBA code.

## 5.4 Ablation Study of Atomic Action Names

To inspect the potential confusion problem stated in Sec. 4.4, we conduct another ablation study for GPT-3.5-Turbo on the full dataset by comparing the impact of adopting the official names and the synonyms. The results in Tab. 3 surprisingly show that using the synonyms increases the Pass@1 and obtains lower A50/A90, which means that the model learns to use the synonyms to generate more correct solutions with fewer steps. This result demonstrates the flexibility of our method: it is possible for users to define their own atomic actions and prompt LLMs to use them.

Table 4: Ablation study of atomic action granularity on the chart-related and format-related tasks. The results show that using an integrated CreateChart action achieves slightly lower A50 and A90 but encounters significantly inferior Exec@1 and Pass@1. Additionally, splitting the SetFormat into finer-grained format-setting actions leads to higher Exec@1 and Pass@1.

| Experiment | Method | Exec@1↑ | Pass@1↑ | A50↓ | A90↓ |
|---|---|---|---|---|---|
| Integrating CreateChart | Ours full | **91.7%** | **43.3%** | 1.25 | 1.67 |
| | Ours + Integrated CreateChart | 79.1% | 37.2% | **1.00** | **1.50** |
| Splitting SetFormat | Ours full | 70.7% | 9.8% | 2.75 | 6.65 |
| | Ours + Split SetFormat | **80.5%** | **12.2%** | **2.00** | **5.60** |

## 5.5 The Influence of LLM Temperature on Task Plan Generation

We evaluate the stability of our method by running the full method three times with temperatures from 0.0 to 1.0. The results in Fig. 4 show that the metrics are stable with slight deviations from the values for temperature=0.0.

## 5.6 Atomic Action at Different Granularity

A natural question is how to determine the granularity of atomic actions, i.e. the number of workbook elements an action manipulates.

To investigate this question, two experiments are conducted: 1) The actions that set chart properties are incorporated into the CreateChart action, and the original separate actions are removed, with the expectation that SheetCopilot will set chart properties when creating charts. 2) In another experiment, SetFormat, which is originally used to set multiple format properties, is split into finer-grained format-setting actions. Please refer to Sec. D.2 for details.

We conduct these two experiments with GPT-3.5-Turbo backend on the chart and format-related tasks. The results in Tab. 4 show that using an integrated CreateChart action to handle chart creation and property setting simultaneously obtains slightly higher efficiency (lower A50 and A90). However, this variant encounters significantly inferior Exec@1 and Pass@1. In contrast, splitting the original SetFormat action witnesses considerable gains in Exec@1 and Pass@1.

After analyzing the results, we found that an integrated CreateChart encounters lower functional correctness as its complex document makes it difficult for SheetCopilot to understand the action usage, thus being less able to correctly determine action arguments. In addition, the lengthy documentation of this integrated action frequently exceeds GPT-3.5's token limit. In contrast, we observed that after splitting SetFormat, the LLM can easily understand the simple finer-grained actions, thereby encountering fewer hallucination cases. See Sec. D.2 for detailed analysis.

These results suggest that it is more desirable to use finer-grained atomic actions instead of integrated high-level actions in terms of functional correctness.

## 6 Conclusion

We propose SheetCopilot, a spreadsheet agent based on the observe-propose-revise-act framework that decomposes a high-level task into step-by-step solutions for software control. To evaluate our agent, we curate a realistic and diverse dataset representing the typical demand of non-expert spreadsheet users. The experimental results show that SheetCopilot can perform spreadsheet manipulation tasks with a high pass rate and acceptable efficiency, outperforming VBA-based methods. The ablation studies show that the closed-loop control and external document insertion used by SheetCopilot bring clear improvements over the baseline and that adopting a set of atomic action names dissimilar to the official names achieves a surprising performance gain. We also find that utilizing finer-grained atomic actions instead of integrated high-level actions can notably improve functional correctness. We hope our work provides a useful roadmap for researchers interested in the field of LLM-based autonomous agents and sheds light on future research.

## Acknowledgments and Disclosure of Funding

This work was supported in part by the National Key R&D Program of China (NO. 2022ZD0160102), the National Natural Science Foundation of China (No. 61836014, No. U21B2042, No. 62072457, No. 62006231), Hong Kong Institute of Science & Innovation, Chinese Academy of Sciences, and the InnoHK Fund.

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

# 7 Supplementary Material

## A Details of Dataset Collection

### A.1 Details of Seed Task Collection

The details of the keyword-based and LLM-based filters described in Section 3.1 are shown below:
**Keyword-based filtering**: Questions containing keywords about irrelevant spreadsheet control tasks (e.g., Visual Basic for Application (VBA), loading and saving files, keyboard and mouse shortcuts, using images and shapes) are removed. **LLM-based filtering**: To easily identify irrelevant question and answer (Q&A) pairs, we introduce the seventh task category, i.e. Invalid, apart from the six categories. With these categories, a set of Q&A pairs are labeled as exemplars to prompt ChatGPT to assign at least one category label to each pair (See Tab. B for the used prompt). After classification, we remove 2432 pairs labeled as Invalid, thereby obtaining a clean dataset containing 13574 pairs annotated with task categories.

We notice that multiple Q&A pairs are likely to refer to similar tasks with different expressions. To identify representative questions, clustering is performed within each combination of task categories (note that a pair may be labeled with multiple task categories). Specifically, the title and text of each question are concatenated as a single paragraph which is then embedded into a 768d vector[2]. For speeding up computation, all embeddings within a category combination are further projected to 64d vectors using Principle Component Analysis. Finally, these embeddings are grouped into 10 clusters using K-means++, which means that 10 representative questions are selected for each category combination. Clustering all category combinations, 220 representatives are selected from the 13574 pairs.

As our evaluation environment does not support excessively complicated tasks (e.g., consolidating data, setting page layout, connecting pivot tables), out of these representatives, we further select 67 that involve operations supported by the evaluation environment. The tasks described in these questions are regarded as the seed tasks that capture the most important patterns of the whole clean dataset.

### A.2 Details of Workbench Collection

To lessen the difficulties of collecting a large-scale task dataset, the evaluation workbooks in the workbench strictly follow three rules: (1) Data in every sheet starts from cell A1; (2) Blank columns within tables are not allowed; (3) Each column is provided with a header on the first row, which means that the records of each column are listed from the second row. To better utilize the internal knowledge of LLMs, we also provide a short paragraph describing the context of each evaluation workbook. This context briefly introduces the usage and domain of the workbook and includes useful formulas that assist in manipulating the workbook. See Tab. A for the contexts of all evaluation workbooks.

### A.3 Details of Core Set Collection

**Adaptation**. During adaptation, GPT-4 is prompted to change the manipulated sheet elements (e.g. ranges, charts, data types) mentioned in the seed tasks. To enable GPT-4 to generate clear task instructions relevant to the evaluation sheets, three types of descriptions are included in the prompt, i.e., a brief sheet summary, column headers, and row numbers. In addition, adaptation exemplars written by the authors are included in the input to teach GPT-4 in a few-shot manner. In the end, a batch of 10 seed tasks to be adapted is listed so that the token limit is fully utilized, which speeds up the adaptation process and saves the cost. See the used prompt in Tab. C and adaptation examples in Tab. D.

**Verification**. Between adaptation and simplification, an extra verification step is required since a number of the adapted tasks are unrealistic and irrelevant to the target sheet. To retain valid proportions, we prompt GPT-3.5 (GPT-4 is not used as it is overly expensive) to classify each task according to four criteria, i.e., realism, clarity, relevance, and completeness. A task is valid only if it simultaneously fulfills the four criteria. GPT-3.5 is asked to describe how each task fulfills the criteria before determining its validity as we empirically find that this method more accurately

---

[2]The model used for embedding is OpenAI text-embedding-ada-002.

Table A: The names and contexts of the collected workbooks. A context describes the usage and domain of the corresponding workbook as well as useful formulas used to manipulate the workbook.

| Workbook Name | Context |
|---|---|
| Invoices | My workbook records many invoices made on different dates. |
| SalesRep | My workbook records the monthly sales of all employees. |
| SummerSales | My workbook records the sales of my company in the summer. |
| EntireSummerSales | My workbook records the sales of my company in the summer. |
| ShippingCosts | My company needs to deliver the goods to customers by truck. My workbook records the distances between my customers and four destinations. The per-mile shipping charge is 3.11 with a minimum charge of 75. |
| EntireShippingCosts | My company needs to deliver the goods to customers by truck. My workbook records the distances between my customers and four destinations. The per-mile shipping charge is 3.5 with a minimum charge of 80. |
| PricingTable | My workbook contains two tables: Sheet "Sheet1" records my transactional data which are the number of rolls of fence sold on certain dates. Sheet "Pricing Table" is a pricing table used to determine the price per roll according to the range the roll number falls in (The range is bounded by Units From and Unit To). |
| BoomerangSales | My workbook has two tables. Sheet "Sheet1" records the sales of a boomerang company. Sheet "Retail Price" lists the retail prices for all products. |
| WeeklySales | My workbook records weekly sales and COGS but the profit has not been calculated. The necessary formula is Profit = Sales - COGS. |
| NetIncome | My workbook records revenue and expense. Net Income = Revenue - Total Expenses. |
| PeriodRate | My workbook records the annual rates of my investments. A year can consist of several periods. |
| Tax | My workbook records the weekly sales of my company and is used to compute taxes. The necessary formulas are as follows: Profit Before Tax = Sales - Total Expenses Before Tax; Tax Expense = Profit Before Tax * Tax Rate. |
| MaturityDate | My workbook records my loans with their lengths in days. |
| StockChange | My workbook records the values of my stocks on two dates. |
| IncomeStatement | My workbook records the yearly accounting data of my company. The necessary accounting formulas are as follows: Gross Profit = Net Sales – Cost of Goods Sold (COGS); Operating Profit = Gross Profit - Operating Expenses; Net Profit = Operating Profit - Tax Expense. |
| IncomeStatement2 | My workbook records the yearly accounting data of my company. The necessary accounting formulas are as follows: Gross Profit = Net Sales – Cost of Goods Sold (COGS); Net sales = Sales - Sales return - Discounts and allowances; Cost of goods sold = Materials charges + Labor charges + Overhead; Gross profit = Net sales - Cost of goods sold. |
| SmallBalanceSheet | My workbook records the total assets, liabilities, and owner's equity. Here are the necessary financial formulas: Assets = Current Assets + Fixed Assets + Other Assets; Liabilities & Owner's Equity = Current Liabilities + Long-term Liabilities + Owner's Equity. |
| SimpleCompoundInterest | My workbook is blank and used to record the interests of my investment. The necessary formulas are as follows: Simple Interest = Principle amount * Year * Interest rate; Compound Interest = Principle amount * (1 + Interest rate) ^Year. |
| FutureValue | My workbook records several investments whose future values need to be calculated according to the formula Future value = Present value * (1 + Annual Interest Rate / # Compound periods) ^(Years * # Compound periods). |
| PresentValue | My workbook records several investments whose present values need to be calculated according to the formula Present value = Future value / (1 + Annual Interest Rate / # Compound periods) ^(Years * # Compound periods). |
| ExpenseReport | My workbook records all aspects of expenses but has not yet been completed. The necessary formulas are as follows: Tax = Subtotal * Tax rate; Total = Subtotal + Tax. |
| DemographicProfile | My workbook records information of respondents. |
| GDPBreakdown | I have two sheets: Sheet "Sheet1" records economic indicators of countries across the years. Sheet "Sheet2" records a list of chosen country names. |
| EasyGDPBreakdown | My workbook records the economic indicators of countries across many years. |
| XYScatterPlot | My sheet shows how two variables (Range and Height) change along with the projection angle. |
| VelocityDisplacement | My sheet records velocity against displacement. |
| Dragging | My sheet records data from an experiment where one hanging block (m2) drags a block (m1=0.75 kg) on a frictionless table via a rope around a frictionless and massless pulley. |
| RampUpAndDown | My sheet records the accelerations of a block in two physical scenarios but has not been completed. One scenario is in columns A to B while the other is in C to D. |

identifies invalid tasks. Finally, 1515 valid tasks (90.7%) remain. See Tab. F for the used prompt and an example of using GPT-3.5 to verify the tasks.

**Simplification**. As the adapted task instructions are likely to refer to specific functions and operations built in Excel, these instructions need to be simplified to read like the tone and fashion of a non-expert user. We establish the following rules for prompting GPT-3.5 to simplify the tasks: 1) Convert specific mentions to natural language descriptions and remove redundant words while retaining the original intention and order of actions. 2) Avoid referring to existing columns by the column indices since it is more natural to refer to a column by the column header. 3) Mention the insertion place and add a column header when inserting a new column to avoid ambiguity. 4) Finally, use domain-specific knowledge to diversify the expression of the generated instructions. See Tab. G for the prompt and an example.

After these steps, a raw dataset containing diverse, natural, and realistic tasks is produced. To extract a core set from the simplified tasks, six random tasks are selected for each category combination, resulting in 402 selected tasks. The authors polish these selected tasks by further revising them and discarding a fraction not supported by the simulator, resulting in 184 remaining tasks. To enrich the core set, the authors act as non-expert users to compose 37 more tasks. Lastly, 221 tasks exist in the core set. See Fig. A for the instruction length and atomic action distributions, Fig. B for the proportions of the six task categories and the task diversity of the core set, and Fig. C for the numbers of each category combination.

The final step is to prepare reference solutions for these tasks. To objectively judge the performance of LLMs, we use GPT-3.5 to generate multiple solutions for each task and then retain the successful ones after verifying the final s by hand. As a number of solutions are likely to exist for a task, multiple reference solutions are collected as the ground truth for one task.

## A.4 Selecting the 10% Datast Used for Comparing the LLMs

20 tasks are selected from the 221-task core set to approximately maintain the percentages of the six categories and the distribution of the numbers of atomic actions. This collection basically represents the pattern of the core set, avoiding a data distribution shift to a large extent. See Fig. D for the statistics of this subset.

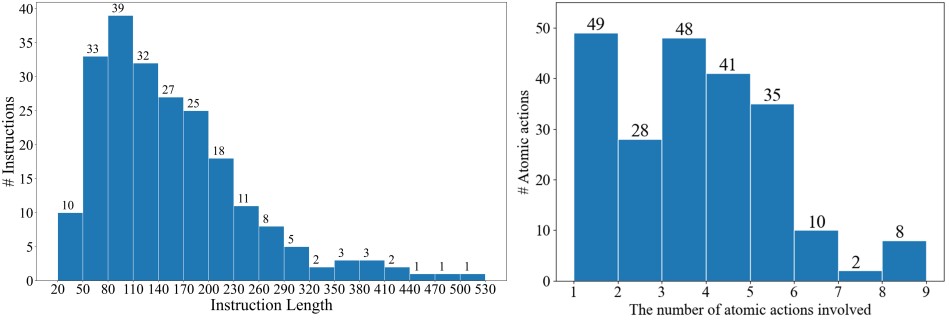

Figure A: The distributions of the instruction lengths and the numbers of atomic actions involved in each instruction. The two histograms demonstrate the diverse task complexity of the core set.

## B Atomic Action Names

### B.1 Collecting Atomic Actions

Selecting a set of indivisible and easy-to-understand atomic actions is necessary since LLMs need to generate an interpretable solution to the user-specified task. As no existing atomic actions used in spreadsheet software have been curated and publicized, we obtain the names of these actions during classifying the SuperUser Q&A pairs (Described in Sec. A.1). Specifically, ChatGPT is prompted to determine which atomic actions each Q&A pair requires to complete the task, which resembles a text summarization problem. As multiple similar names are likely to be generated for an atomic action, we

Table B: An example of using the classification prompt to predict multiple category labels and extract the involved atomic actions for the Q&A pairs scraped from SuperUser. For clarity, the prompt components and an example ChatGPT response are marked with brief descriptions in blue.

(Requirements and category definitions for classification)
I have a batch of Excel questions for you to clean. I need you to determine which categories each question belongs to according to the question title, and its top answer. A question may belong to multiple categories if it involves the atomic actions of these categories.
The six categories are defined as below:
A: Entry and Manipulation. Tasks that enter and manipulate cell contents such as editing cell values, splitting texts, inserting and deleting rows/columns.
B: Management. Tasks that organize data such as sorting, filtering, using slicers to filter data, and moving rows/columns.
C: Formatting. Tasks that modify sheet or cell formats such as changing text font, adding drop-down lists, and merging cells.
D: Formula. Tasks related to using formulas such as performing calculations and statistical analysis.
E: Charts. Tasks related to presenting data visually such as creating charts, customizing chart elements, and adding a trend line.
F: Pivot Table. Tasks related to using Pivot Table such as creating and formatting Pivot Tables.
If the question does not belong to any of the six categories above or it is too vague, it is regarded as "Invalid".
I also need you to determine which atomic actions each valid question requires to complete the task.

(Classification exemplars collected by the authors.)
Below are example questions in the form of a Python dictionary:
Questions:
Q1: {"question title":"How to filter a cell which containing data by other list", "top answer":"You have to select the Column Headers and click on "Data", "Filter" and "automatic filter" for XL2003
"Data", "Filter" for XL2007 and probably 2010."}
Q2: {"question title":"Group rows by column then sort by another column", "top answer":"I found the solution myself. First sort by the ID then I added a new column with the following formula:=INDEX($B$1:$B$278,MIN(IF($I$2:$I$278=$I4,ROW($I$2:$I$278)))). Replace the bounds as necessary. I then sort by this new column and then the ID column and the data is in the order I wanted."}
Q3: {"question title":"How to delete the infinite blank rows?", "top answer":"Go to Home tab → Find and Select → Go to Special Select Blanks. Click OK. The blank cells will be highlighted... "}
Q4: {"question title":"How to cut certain rows and insert before a certain row in excel", "top answer":"Highlight the row by clicking of the row number then press Ctrl+X then click on the row number below the row were you intend to insert the cut rows then press Ctrl+V."}
Q5: {"question title":"How do I add conditional formatting to cells containing #N/A in Excel?", "top answer":"Select the first cell you want to highlight. (B6).
Click Home -> Conditional Formatting -> Manage Rules -> New Rule. Select Use a formula to determine which cells to format. In the field Format values where this formula is true, enter =ISNA($B6). Click Format to set the cell formatting, then select OK .
Click OK again to create the formatting rule. In the Conditional Formatting Rules Manager, edit the range under Applies to (ex: $B6:$B8). Select OK to apply the rule, which will match to true and thus apply the formatting you want."}
Q6: {"question title":"How do you auto resize cells in Excel?", "top answer":"I have solved it like this: 1. Select the entire spreadsheet. 2. Double click the line in the margin in between the two rows. This auto sizes every row not just the one row that I clicked on."}
Q7: {"question title":"Sum contents in column from starting cell on down without setting an explicit last cell index", "top answer":"For Excel 2003 or before: =SUM(B5:INDEX(B5:B65536,MATCH(TRUE,INDEX(ISBLANK(B5:B65536),0,0),0)-1,0)). For Excel 2007 or after: =SUM(B5:INDEX(B5:B1048576,MATCH(TRUE,INDEX(ISBLANK(B5:B1048576),0,0),0)-1,0))"}
Q8: {"question title":"How to get the dates of the current monday, wednesday and friday of the current week in excel?", "top answer":"The following will display the dates for Monday, Wednesday and Friday of the current week:
=NOW() - WEEKDAY(NOW(),3), =NOW() - WEEKDAY(NOW(),3)+2, =NOW() - WEEKDAY(NOW(),3)+4
Basically this is taking the time now, and subtracting the current weekday (which gives you Monday), then adds 2 days or 4 days to get Wednesday and Friday."}
Q9: {"question title":"Excel: how to adjust range in pivot table automatically?", "top answer":"First, turn your data table into an Excel Table object: Select any cell in the source range, then click Insert >Table or keyboard shortcut Ctrl-T..."}
Q10: {"question title":"Create a pie chart from distinct values in one column by grouping data in Excel", "top answer":"This is how I do it:
1. Add a column and fill it with 1 (name it Count for example). 2. Select your data (both columns) and create a Pivot Table: On the Insert tab click on the PivotTable | Pivot Table (you can create it on the same worksheet or on a new sheet) ..."}
Q11: {"question title":"How to Change Orientation of Multi-Level Labels in a Vertical Excel Chart?", "top answer":"You can only control the alignment of the inner most set of multi-level axis labels. Except when you add a data table to the chart, then you have no
control over the alignment. One thing you can consider is to turn off the multi-level category option:"}
Q12: {"question title":"Create PDF with internal hyperlinks", "top answer":""}
Q13: {"question title":"What does the TABLE function do?", "top answer":""}
Q14: {"question title":"How do you create a report in Word (or other documentation software) that is linked directly with Excel 2007 data?", "top answer":""}
(Exemplar classification results consist of the category labels and involved atomic actions of each Q&A pair)
Results:
Q1: {"category":"B", "atomic actions":"Filter"}
Q2: {"category":"B, D", "atomic actions":"Sort, Statistic functions, Lookup and reference functions"}
Q3: {"category":"A", "atomic actions":"Delete rows"}
Q4: {"category":"A", "atomic actions":"Cut-paste, Insert rows"}
Q5: {"category":"C", "atomic actions":"Conditional formatting"}
Q6: {"category":"C", "atomic actions":"Resize cells"}
Q7: {"category":"D", "atomic actions":"Math functions"}
Q8: {"category":"D", "atomic actions":"Date and time functions"}
Q9: {"category":"F, A", "atomic actions":"Create Pivot Tables, Insert rows"}
Q10: {"category":"A, F, E", "atomic actions":"Insert columns, Create Pivot Tables, Create Pivot Charts"}
Q11: {"category":"E", "atomic actions":"Change axis label"}
Q12: {"category":"Invalid", "atomic actions":None}
Q13: {"category":"Invalid", "atomic actions":None}
Q14: {"category":"Invalid", "atomic actions":None}

(Q&A pairs to be classified)
Please process the questions below and give results in the same form as the examples above.
Questions:
Q1: {"question title":"SUM column C unless value in col A matches col B", "top answer":"You can use the following, entered as an array formula with CTRL+SHIFT+ENTER:=SUM(IF(COUNTIF(B1:B7,A1:A7)=0,C1:C7))"}
Q2: {"question title":"Word paste table from Excel has marks instead of values", "top answer":"Cause: The column is "too narrow" to allow showing the content, according to Word. Remedy: Move the mouse pointer slowly horizontally over the column border and it will change shape at some point ..."}
Q3: {"question title":"According to 4^3^2 = (4^3)^2. Is this really the standard mathematical convention for the order of exponentiation?", "top answer":"Your teacher is right. According to standard rules of algebra exponents are solved from right to left, therefore ..."}
...

(GPT-3.5 response)
Q1: {"category":"A, D", "atomic actions":"Update cell value, Math functions"}
Q2: {"category":"C", "atomic actions":"Formatting cells"}
Q3: {"category":"Invalid", "atomic actions":None}
...

Table C: An example of using the adaptation prompt to adapt the seed tasks. For clarity, the prompt components and an example GPT-4 response are marked with brief descriptions in blue.

---

(Requirements for adaptation)
As an Excel expert, you have been assigned the responsibility of adapting a set of task instructions for specific Excel workbooks. These instructions will be utilized to evaluate the Excel manipulation capabilities of large language models.

Requirements:
1. First, identify individual atomic actions used in the original instructions, then develop new instructions incorporating these actions.
2. Use the detailed descriptions of the provided workbooks to modify the original instructions so that they become compatible with the workbooks. Specifically, you must change the manipulated objects (ranges, sheets, rows, and columns) in the original instructions. You must also change the way you use atomic actions. For instance, if the original instruction sets the data type as accounting, you can change it to other types in the adaptation.
3. Use standard range references, such as 'Sheet2!A1:C9', 'A2:E16', 'A:H', column C, or row 16.
4. Use different phrasing (e.g., various sentence structures and noun/verb forms) to create diverse instructions.
5. Apply domain-specific knowledge to diversify the generated instructions. For instance, use financial knowledge to calculate various metrics, demonstrate trends in product sales from different perspectives, visualize data using various types of charts, and so on.
6. (Important!) The generated instructions must describe realistic tasks and involve the normal use of atomic actions according to the workbook descriptions.
7. In every new instruction, new tables and Pivot tables must be created in a new worksheet and start from A1. Only one new sheet is allowed to be created. Besides, the headers of new columns/rows and the names of new sheets must be set.
8. The instructions after adaptation should look like what a non-expert Excel user would say and should not mention any specific functions or operations built in Excel.
(Available atomic actions used to label the adaptation results)
Here are the atomic actions you can identify within the six categories:
A. Entry and manipulation: Update cell value, Delete, Split text to columns, Insert row, Insert column, Autofill, Copy-paste, Copy-paste format, Copy sheet, Cut-paste, Find and replace, Set hyperlink, Delete hyperlink, Remove duplicates, Rename sheets, Insert checkbox, Insert textbox, Create sheet, Delete sheet, Clear
B. Management: Switch sheet, Sort, Filter, Delete filter, Slicer, Move rows, Move columns, Group, Ungroup, Hide rows, Hide columns, Unhide rows, Unhide columns, Hide sheet, Unhide sheet, Set password, Transpose, Create named range, Delete named range, Data consolidation, Freeze panes, Unfreeze panes, Split panes
C. Formatting: Format cells, Set data type, Delete format, Merge cells, Unmerge, Change page layout, Set border, Resize cells, Conditional formatting, Lock and unlock, Protect, Unprotect, Drop-down list, Data validation, Insert checkbox, Display formulas, Wrap text, Unwrap text, Autofit
D. Charts: Create chart, Create Pivot Chart, Set chart title, Set chart axis, Set chart has axis, Set chart legend, Set chart type, Set chart color, Set chart source, Set chart marker, Resize chart, Set trend line, Add data labels, ... (leave out for clarity)
E. Pivot Table: Create Pivot Table, Remove Pivot Table, Set summary type, Sort Pivot Table
F. Formula: Date and time functions, Logical functions, Lookup and reference functions, Math functions, Statistical functions, Text functions, Financial functions

Restrictions for the atomic action parameters:
Chart type can only be (stacked/clustered) column chart, (stacked/clustered) bar chart, (3D) pie chart, line chart (with smooth lines), (stacked) area chart, or scatter chart.
Cell value type can only be date, text, time, currency, percentage, number, or general.

I will give you an example first:

(Adaptation exemplars written by the authors)
Given an Excel workbook:
The sheet 'Sheet1' records the employee working hours of a coffee shop. It has 8 columns (Headers are: A: "location", B: "name", C: "date", D: "hours", E: "ot hours", F: "base pay", G: "ot pay", H: "total pay") and 11 rows (including the header row). The cells in the "location" column can be "capitol hill", "queen anne". The cells in the "name" column can be "Aaron", "Bob", "Blanca", "Frank".

The original instructions to be adapted:
1. I'd like to know if there's a formula (or combination of formulas) to sum Column B ONLY if it equals Column A?
2. Column A contains multiple due dates and cell B1 is 10/31/2022. Append a special character to the cells in column A depending on whether the due date is less or more than the date in B1. If neither applies, leave the cell unchanged.
3. Create a Pivot Table to separate dates by each month similar to the quarterly function. This will show all dates in the last year under the column label. Choose the monthly option under data filters in the columns label under data filters.
4. Freeze A1:B10 so that no matter how I scroll vertically or horizontally this range is always frozen.
5. I have five groups of data each occupying two columns. I'd like to have them plotted all on one bar chart but with the series separated (i.e., not clustered) so that several column charts stick together sharing the same y-axis.

Adaptations compatible with the given workbook (Show the categories involved in the generated instruction and list the atomic actions following the category label):
1. Instruction: In a new column with header "Total pay each location", use a formula to sum the "total pay" (Column H) for each employee ONLY if their "location" (Column A) is "capitol hill". - Categories (atomic actions): A (Update cell value); F (Math functions)
2. Instruction: Create a formula in a new column (Column I) with header "Marked Due Dates" to check if the dates in column C are less or more than 10/31/2022. If the date is less, append a '-' to the cell in the same row in column C; if the date is more, append a '+'; otherwise, leave the cell unchanged. - Categories (atomic actions): A (Update cell value, Autofill); F (Logical functions, Text functions)
3. Instruction: Create a Pivot Table in a new sheet named "Sheet2" based on the data in 'Sheet1' and then summarize sum of hours for each location in this Pivot Table. - Categories (atomic actions): A (Create sheet); E (Create Pivot Table)
4. Instruction: Freeze the range A1:H1 so that the headers remain visible when scrolling vertically or horizontally. - Categories (atomic actions): C (Freeze panes)
5. Instruction: Create a Pivot Table in a new sheet named "Sheet2" to sum the hours for all dates and then plot a line chart to show the trend of hours changing with dates. - Categories (atomic actions): A (Create sheet); E (Create Pivot Table); D (Create chart)

(Seed tasks to be adapted)
Now it's your turn.
Given an Excel workbook:
My workbook records information of respondents. Sheet "Sheet1" has 4 columns (Headers are A: "Respondents", B: "Sex", C: "Civil Status", D: "Highest Educational Attainment") and 41 rows (including the header row). The cells in the "Respondents" column range from 1.00 to 40.00. The cells in the "Sex" column can be "Male", "Female". The cells in the "Civil Status" column can be "Single", "Widow", "Married". The cells in the "Highest Educational Attainment" column can be "Elementary", "Senior High School", "Junior High School".

The original instructions to be adapted:
1. I'd like to know if there's a formula (or combination of formulas) to sum Column B ONLY if it equals Column A.
2. I have a sheet 1 with 3 columns as shown below. I have another sheet named Sheet 2 with only the first names in Column A in Sheet 1. I would like to write a formula in column B of Sheet 2 which matches the first name in Column A of Sheet 2 against the first name in Column A of Sheet 1. If "no" is found in column C of Sheet 1 then copy the information in column B of Sheet 1 to column B of Sheet 2.
...
Adaptations compatible with the given workbook (Show the categories involved in the generated instruction and list the atomic actions following the category label):
(GPT-4 response)
Instruction: Can we find out if there's a formula (or a combination of them) that sums up all values in the "Highest Educational Attainment" column (Column D) ONLY if the "Sex" is "Male" (Column B)? - Categories (atomic actions): A (Update cell value); F (Math functions, Logical functions)
Instruction: In a new column (Column E) named "Gender-Match Education", find if the "Sex" in column B matches the "Highest Educational Attainment" in Column D. If there's a match, copy the corresponding "Civil Status" from Column C to Column E. - Categories (atomic actions): A (Insert column, Update cell value, Autofill); F (Lookup and reference functions)
...

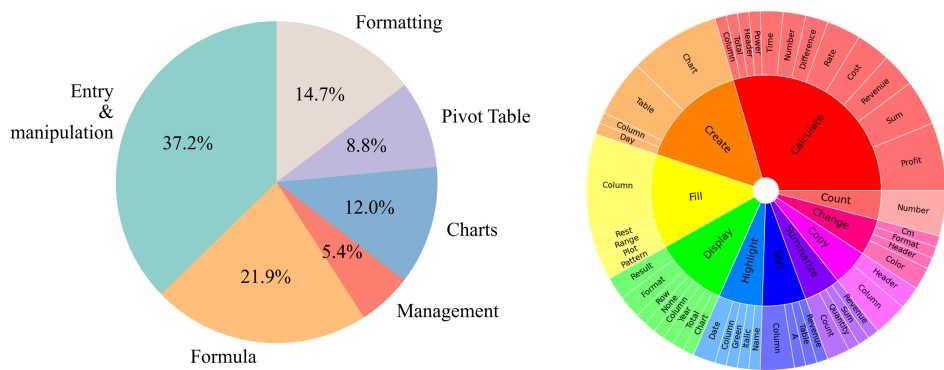

Figure B: Left: The proportions of the six categories. Right: Diversity of the verb-noun phrases in the core set. We demonstrate the diversity of the core set by showing the top 10 most frequent root verbs (the inner circle) and their direct noun objects (the outer circle) in the instructions.

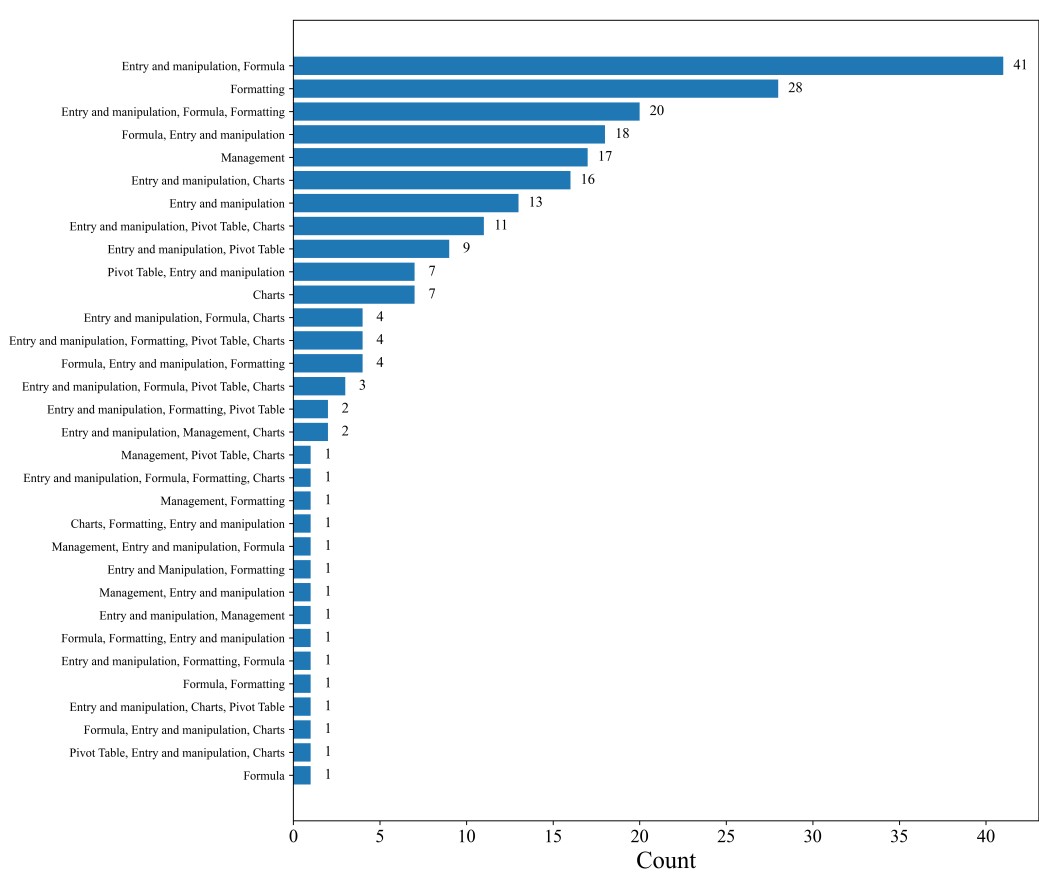

Figure C: The distribution of the category combinations.

Table D: **Adaptation examples**. The adaptations show that using the proposed prompt, GPT-4 is able to adapt the seed tasks for the evaluation sheet by changing the operated sheet elements and use of the atomic actions.

| Seed tasks | Adaptations |
|---|---|
| • Count the number of values that appear more than once in column A. | • In a new cell, count the number of "Product" (Column C) values in 'Sheet1' that appear more than once. |
| • Add leading zeros to every number in column A then append them to the texts in column B. | • In a new sheet named "Sheet2", format the years in column A from "Sheet1" as text and append the corresponding Gross Profit value (Column J) from "Sheet1". |
| • Apply conditional formatting in a column based on two other columns. If the cell value in column C is not 'End' and the cell value in column D is empty, the value in column A is displayed in red. | • Apply conditional formatting to cells in column A based on the cell value in column B. If the value in A is greater than the value in B, display the value in column A in red. |
| • I have five groups of data each occupying two columns. I'd like to have them plotted all on one bar chart but with the series separated so that several column charts stuck together that share the same y-axis. | • Create a Pivot Table in a new sheet named "Sheet2" to show the sum of distances for each customer to each destination, and then create a clustered bar chart to represent the distances for each destination on the same y-axis. |

Table E: **Simplification examples**. The simplification results demonstrate three features: (1) using brief and clear references to operated ranges; (2) expressing the intention with spoken language instead of directly mentioning Excel built-in functions; (3) considerably reducing the sentence length.

| Adaptations | Simplification Results |
|---|---|
| • Calculate the sum of "Quantity" (Column E) in 'Sheet1' ONLY if the "Product" (Column C) in 'Sheet1' matches the "Product" (Column A) in 'Retail Price' sheet. | • Find the total quantity sold for each product listed in the 'Retail Price' sheet. |
| • In a new sheet named "Sheet3", use INDEX and MATCH to retrieve sales information from "Sheet1" based on the row IDs provided in a list on another sheet. | • Retrieve sales information based on row IDs listed on another sheet and display the results on a new sheet named "Sheet3". |
| • In a column (Column H) with header "Days from Today", calculate the number of days difference between the "Date" (Column A) and today's date. | • Calculate the number of days between today and the "Date" column in a new column named "Days from Today." |

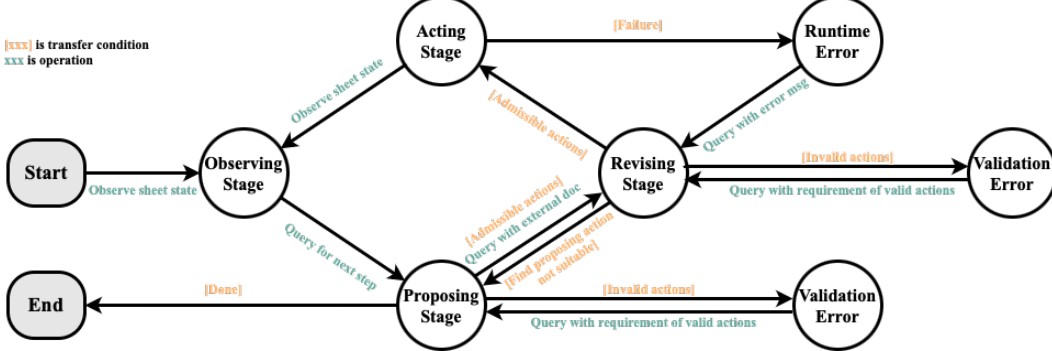

Figure D: Statistics of the subset used for comparing the LLMs.

Figure E: The complete state machine for task planning. The transfer condition (orange texts with brackets) must be fulfilled before performing the corresponding operation (green texts) to transfer to a next state.

Table F: An example of using the verification prompt to classify the adapted tasks. For clarity, the prompt components and the GPT-3.5 response are marked with brief descriptions in blue and the redundant contents are substituted with ellipsis.

I will give you a batch of Excel task instructions that are utilized to evaluate the spreadsheet manipulation capabilities of large language models. Please check all instructions according to the criteria and the descriptions of the given workbooks.

Requirements:

1. If an instruction fulfills all of the following four criteria strictly at the same time, it is valid.
A. Realism: Verify that the instructions are representative of real-world Excel problems encountered by expert users. Avoid including contrived or overly complex tasks that would rarely be encountered in practice.
B. Relevance: Verify that the instructions are relevant to the context of the given workbooks. Reject instructions that do not relate to the content of the workbooks or are not applicable in the context of Excel.
C. Clarity: Verify that the instructions are clear and easy to understand. They should be free from grammatical errors and ambiguities.
D. Completeness: Verify that the instructions can be completed using the provided workbook data and Excel features. If additional data or functionality is needed to accomplish a task, reject the instruction.

2. Use your domain-specific knowledge to reason and make decisions. For example, it is unlikely to calculate monthly or yearly averages of some physical values because this task is impractical.

(Exemplars for verification. The sheet description is provided to more accurately verify the fulfillment of the four criteria)
I will give you an example first:

Given an Excel workbook:
My workbook records many invoices made on different dates. Sheet "Sheet1" has 7 columns (Headers are A: "No.", B: "Date", C: "Salesman", D: "Product", E: "Price", F: "Units", G: "Sales") and 19 rows (including the header row). The cells in the "No." column range from 10500.00 to 10505.00. The cells in the "Date" column can be "2011-05-28 00:00:00", "2011-05-25 00:00:00", "2011-05-27 00:00:00". The cells in the "Salesman" column can be "Joe", "Chin", "Moe". The cells in the "Product" column can be "Quad", "Majestic", "Bellen", "Carlota", "Alpine". The cells in the "Price" column range from 22.00 to 32.00. The cells in the "Units" column range from 4.00 to 25.00. The cells in the "Sales" column range from 128.00 to 750.00.

Instructions to check:
1. Find the sales value corresponding to each No. in a new column called "Invoice Lookup".
2. Compare the names in the Salesman column to find the closest match for each name. Put the results in a new column named "Salesman Matched".
3. Create a sheet named "Sheet2" to summarize the total sales for each sales representative in Sheet1.
4. Prepend leading zeros to each number in the "No." column so that they have a fixed length of 6 digits. Append the new results to the corresponding product names in the "Product" column, and put the results in a new column named "Padded No.".
5. Find the corresponding date for each No. in a new column named "Lookup Date" in Sheet1.
6. Find and display the row values in Sheet1 based on the "No." column. Put the results in a new sheet named "Sheet2".
7. Round the values in the "Sales" column to two decimal places in a new column named "Rounded Sales", and display the results with trailing zeros.
8. Merge cells A1 through C1 with cells A2 through D2.
9. Add hyperlinks to each cell in the "No." column that link to its corresponding file.

Check results (Give brief reasons in the comments and tell me if the instruction is valid):
1. Realism: Yes, Relevance: Yes, Clarity: Yes, Completeness: Yes. Comment: The instruction fulfills the 4 criteria, so it is valid.
2. Realism: No, Relevance: No, Clarity: No, Completeness: No. Comment: This instruction is unrealistic and unclear and does not seem to be relevant to the context of the given workbook, so it is invalid.
3. Realism: Yes, Relevance: Yes, Clarity: Yes, Completeness: Yes. Comment: The instruction fulfills the 4 criteria, so it is valid.
4. Realism: No, Relevance: Yes, Clarity: Yes, Completeness: Yes. Comment: The instruction appends numbers to product names, which is not a realistic requirement, so it is invalid.
5. Realism: Yes, Relevance: Yes, Clarity: Yes, Completeness: Yes. Comment: The instruction fulfills the 4 criteria, so it is valid.
6. Realism: Yes, Relevance: Yes, Clarity: Yes, Completeness: No. Comment: The instruction does not specify what values to display, so it is invalid.
7. Realism: Yes, Relevance: Yes, Clarity: Yes, Completeness: Yes. Comment: The instruction fulfills the 4 criteria, so it is valid.
8. Realism: No, Relevance: No, Clarity: Yes, Completeness: Yes. Comment: The instruction merges cells, which destroys the original data and is meaningless in the context of the workbook, so it is invalid.
9. Realism: Yes, Relevance: Yes, Clarity: Yes, Completeness: No. Comment: The instruction does not refer to specific corresponding files, which is incomplete, so it is invalid.

(Instructions to be verified)
Now it's your turn.

Given an Excel workbook:
My workbook records all aspects of expenses but has not yet been completed. The necessary formulas are as follows: Tax = Subtotal * Tax rate; Total = Subtotal + Tax. Sheet "Sheet1" has 6 columns (Headers are A: "Date", B: "Vendor/Client", C: "Expense Account", D: "Subtotal", E: "Tax", F: "Total") and 25 rows (including the header row). The cells in the "Date" column can be "2020-01-06 00:00:00", "2020-01-08 00:00:00", "2019-12-27 00:00:00", .... The cells in the "Vendor/Client" column can be "Company A", "Company B", "Client B", "Client A". The cells in the "Expense Account" column can be "Material Purchase", "Meals", "Hiring", "Gas", "Car Repairs", "Reception", "Accommodation", "Flight tickets". The cells in the "Subtotal" column range from 127.00 to 1954.00.

Instructions to check:
1. Calculate the sum of the Subtotal column only for the rows that have "Company A" in the Vendor/Client column.
2. For each row in Sheet2, find its matching row in Sheet1, and if the Expense Account is "Accommodation", copy the Subtotal to column B of Sheet2.
3. Calculate the tax by multiplying the Subtotal column by 0.1 and write the results in a new column with the header "Tax Calculation".
4. Calculate the number of days between each date in the Date column and the current date. Write the results in a new column with the header "Days Since Expense".
5. Count the number of Vendor/Client values that appear more than once in the data.
6. For each row, subtract the Expense Account's first three characters from the Subtotal and put the result in a new column named "Adjusted Subtotal"
...
Check results (Give brief reasons in the comments and tell me if the instruction is valid):

(GPT-3.5 response)
1. Realism: Yes, Relevance: Yes, Clarity: Yes, Completeness: Yes. Comment: The instruction fulfills the 4 criteria, so it is valid.
2. Realism: Yes, Relevance: Yes, Clarity: Yes, Completeness: Yes. Comment: The instruction specifies a matching condition and copying of data based on that condition, so it is valid.
3. Realism: Yes, Relevance: Yes, Clarity: Yes, Completeness: Yes. Comment: The instruction calculates tax based on the Subtotal column, so it is valid.
4. Realism: Yes, Relevance: Yes, Clarity: Yes, Completeness: Yes. Comment: The instruction calculates the number of days between dates, which is a practical task, so it is valid.
5. Realism: Yes, Relevance: Yes, Clarity: Yes, Completeness: Yes. Comment: The instruction requires counting the number of duplicate values, which is a common task, so it is valid.
6. Realism: Yes, Relevance: Yes, Clarity: Yes, Completeness: Yes. Comment: The instruction specifies a calculation based on the Expense Account column, so it is valid.
...

Table G: An example of using the simplification prompt to rephrase overly specific task instructions. For clarity, the prompt components and an example ChatGPT response are marked with brief descriptions in blue and the redundant contents are substituted with ellipsis.

---

(Requirements for simplification)
You have been tasked with paraphrasing a set of instructions for Excel tasks.
Requirements:
1. Paraphrase each given instruction so that it is more like what a non-expert Excel user would say while retaining the original intention and order of actions in the instruction.
2. Do not mention any Excel built-in functions in the paraphrases. If the original instruction mentions the names of specific Excel built-in functions, you should replace these functions with spoken language. For example, "Use CONCATENATE to combine cells" mentions "CONCATENATE" and should be rephrased as "concatenate cells". "Use conditional formatting to change the background color of cells" mentions "conditional formatting" and should be rephrased as "Set cell background color if ...". "Write a formula to copy data" should be rephrased as "Copy data".
3. Do not refer to existing columns by their indices (e.g., column A is not desired); instead, mention columns by their headers. For instance, 'In column A' should be rephrased as 'In the Year column'.
4. Do not add column references in brackets. For instance, "subtracting Total Expenses (Column B) from Revenue (Column A)" should be rephrased as "Subtract Total Expenses from Revenue".
5. When inserting new columns, the column indices must be given to avoid ambiguity. Besides, each new column must be given a header, so you need to keep the column headers mentioned in the original instructions.
6. Use domain-specific knowledge to diversify the expression of the generated instructions and do not use the same expression repeatedly.

(Exemplars written by the authors for in-context learning)
I will give you some examples first:
Original instructions:
1. Convert the "Year" column values (Column A) from the format of yyyy.00 to yyyy.
2. In a new column with the header "Combined data", use the CONCATENATE function to combine the data from columns A through Z in each row, including the headers, and then autofill the formula to the last row of data.
3. Use advanced filters in "Sheet1" to display rows with sales data for a specific month (assuming weeks starting from the 1st of each month). For example, filter to show rows with weeks "Week 1" to "Week 4" for the first month.
4. Create a line chart on a new sheet named "Sheet2" with weeks (Column A) on the x-axis and sales (Column B) as the y-axis.
5. Apply conditional formatting to Column H (Net Profit) based on Columns F (Operating Profit) and G (Tax Expense). If the cell value in Column G is not 0 and Column F is greater than 0, display the value in Column H in red.
6. In a new sheet named "Sheet3", use the VLOOKUP function to match each row in "Year" (Column A) in "Sheet1" with the corresponding value in Column B ("Net Sales") from "Sheet1".
7. In a new column (Column G) with the header "Tax Calculation", use a formula to calculate the tax by multiplying the corresponding "Subtotal" (Column D) by 0.1.
Think about the flaws in the original instructions before paraphrasing:
1.
Think: The original instruction refers to columns by indices in brackets, which is undesired.
Paraphrase: Convert the "Year" column format from yyyy.00 to yyyy.
2.
Think: The original instruction mentions Excel built-in functions CONCATENATE, which is undesired.
Paraphrase: Concatenate the data from columns A through Z for all rows and write the results in a new column named "Combined data".
3.
Think: The original instruction mentions an Excel built-in operation (i.e., advanced filters), which is undesired.
Paraphrase: Display only the rows with weeks from Week 1 to Week 4.
4.
Think: The original instruction refers to columns by indices in brackets, which is undesired.
Paraphrase: Create a line chart on a new sheet with the Week column on the x-axis and the Sales column as the y-axis.
5.
Think: The original instruction mentions Excel built-in functions (i.e., conditional formatting) and refers to columns by indices, which is undesired.
Paraphrase: If the cell value in Tax Expense Column is not 0 and that in Operating Profit Column > 0, display the cell text in the Net Profit column in red.
6.
Think: The original instruction mentions Excel built-in functions (i.e., VLOOKUP) and refers to columns by indices in brackets, which is undesired.
Paraphrase: Match cells in the Year column and return the corresponding values in the Net Sales Column. Put the results in a new sheet.
7.
Think: The original instruction refers to columns by indices in brackets, which is undesired.
Paraphrase: Calculate the tax by multiplying the "Subtotal" column by 0.1 in column G named "Tax Calculation".

(Task instructions to be simplified. The sheet description is provided to maintain the relevance)
Now it's your turn. Please follow the requirements to paraphrase the original instructions according to the given workbook descriptions.
Given an Excel workbook:
My workbook records all aspects of expenses but has not yet been completed. The necessary formulas are as follows: Tax = Subtotal * Tax rate; Total = Subtotal + Tax. Sheet "Sheet1" has 6 columns (Headers are A: "Date", B: "Vendor/Client", C: "Expense Account", D: "Subtotal", E: "Tax", F: "Total") and 25 rows (including the header row). The cells in the "Date" column can be "2020-01-08 00:00:00", "2020-01-02 00:00:00", "2019-12-31 00:00:00", ... . The cells in the "Vendor/Client" column can be "Client A", "Client B", "Company A", "Company B". The cells in the "Expense Account" column can be "Reception", "Car Repairs", "Accommodation", "Material Purchase", "Gas", "Meals", "Flight tickets", "Hiring". The cells in the "Subtotal" column range from 127.00 to 1954.00.
Original instructions:
1. Calculate the sum of the "Subtotal" column (Column D) ONLY if the "Vendor/Client" (Column B) equals "Company A".
2. In a new sheet named "Sheet2", write a formula in column B to match the "Vendor/Client" in Column A of "Sheet2" against the "Vendor/Client" in Column B of "Sheet1". If the "Expense Account" (Column C) in "Sheet1" is "Accommodation", copy the "Subtotal" (Column D) from "Sheet1" to Column B of "Sheet2".
3. In a new column (Column G) with the header "Tax Calculation", calculate the tax by multiplying the corresponding "Subtotal" (Column D) by 0.1.
...

Think about the flaws in the original instructions before paraphrasing:

(GPT3.5 response)
1.
Think: The original instruction refers to columns by indices in brackets, which is undesired.Flaw: The original instruction mentions the specific value "Company A" for the "Vendor/Client" column, which is undesired.
Paraphrase: Calculate the sum of the "Subtotal" column if the "Vendor/Client" is for "Company A".
2.
Think: The original instruction mentions Excel built-in functions (i.e., MATCH) and refers to columns by indices in brackets, which is undesired.
Paraphrase: In a new sheet named "Sheet2", find the corresponding "Vendor/Client" in "Sheet1" for each row and copy the "Subtotal" from "Sheet1" to a new column in "Sheet2" if the "Expense Account" is "Accommodation".
3.
Think: The original instruction refers to columns by indices in brackets, which is undesired.
Paraphrase: Calculate the tax by multiplying the "Subtotal" column by 0.1 in a new column named "Tax Calculation".
...

---

Table H: The official names and synonyms of the atomic actions used in our method. The Formula category contains the types of formulas used for calculation in the core set tasks and does not represent specific operations.

| Category | Official Name | Synonym |
|---|---|---|
| Entry and manipulation | Write | RangeInputValue |
| | Delete | DiscardRange |
| | InsertRow | NewRowAtIndex |
| | InsertColumn | ColumnCreation |
| | AutoFill | RangeValueTransfer |
| | CopyPaste | ReplicateRange |
| | FindReplace | AlterRange |
| | SetHyperlink | LinkRangeAssociator |
| | RemoveDuplicate | DistinctData |
| | CreateSheet | WorksheetCreation |
| | Clear | EraseRangeContents |
| Management | Sort | AdvancedRangeSort |
| | Filter | SmartRangeSelector |
| | CreateNamedRange | SetRangeName |
| | FreezePanes | LockRowsColumns |
| Formatting | SetFormat | CustomizeFont |
| | SetDataType | RangeTypeFormatter |
| | Merge | ConcatenateCells |
| | ResizeRowColumn | RangeDimensionAdjuster |
| | SetConditionalFormat | FormatWithRules |
| | SetCellLock | ProtectActiveSheet |
| Charts | CreateChart | GraphConstructor |
| | CreateChartFromPivotTable | CreatePivotGraph |
| | SetChartTitle | ChartTitleSettings |
| | SetChartAxis | CustomizeAxisAttributes |
| | SetChartHasAxis | AxisDisplayManager |
| | SetChartLegend | LegendConfiguration |
| | SetChartType | ChartTypeSwitch |
| | SetChartMarker | DefineSeriesMarker |
| | SetChartTrendline | TrendlineAdjustments |
| | AddDataLabels | DisplayChartDataLabels |
| | AddErrorBars | ErrorBarsIntegration |
| Pivot Table | CreatePivotTable | PivotTableConstructor |
| | SetPivotTableSummaryFunction | PivotFunctionChange |
| Formula | Date and time functions | N/A |
| | Logical functions | |
| | Lookup and reference functions | |
| | Math functions | |
| | Statistical functions | |
| | Text functions | |
| | Financial functions | |

project all generated action names into 768d embeddings using the OpenAI embedding model[3] and then reduce these embeddings to 64d vectors before performing Kmeans++ clustering. This process produces 80 clustering centers each of which represents a group of semantically similar action names. After checking the names corresponding to these centers by hand, we find that these names are the official names used for the built-in features of Microsoft Excel. According to the potential use of atomic actions in the core set tasks and the implementation restrictions of the evaluation environment, a proportion of these action names (44 out of 80) are used by our SheetCopilot (Listed in the initial prompt and used in the external document). These 44 atomic action names are used for conducting the comparison experiment (Sec. 5.2), state machine ablation study (Sec. 5.3), and the stability test experiment (Sec. 5.5). Please see Tab. H for these atomic action names.

## B.2  Collecting Synonyms for the Official Atomic Action Names

To investigate the confusion issue mentioned in Sec. 4.4, we attempt to adopt a different set of atomic action names. Specifically, we use GPT-4 to generate 10 candidate synonyms for each atomic action name according to the low-level implementation of the action. Then, we adopt as the new name the candidate farthest from the official name in terms of the cosine similarity in the embedding space[4]. These synonyms (shown in Tab. H) are used in the ablation study in Sec. 5.4 to investigate the impact of adopting different sets of names.

## C  Details of SheetCopilot Implementation

We introduce the implementation of the state machine our SheetCopilot is based on and the prompt used to query LLMs to generate task solutions.

## C.1  State Machine Implementation

The Observe-Propose-Revise-Act framework is implemented using a state machine shown in Fig. E. At the beginning of each step, the Observing stage enables the LLM to receive the sheet state before planning an action in the Proposing stage. Since the LLM is likely to output an incorrect action, an external document of the initially planned action is inserted into the query to prompt the LLM to revise the action. If the initially planned action is invalid, a validation error will occur and then the state machine will return to the Proposing stage; If the revised action is invalid, a validation error will also occur but the state machine will return to the Revising stage. If the revised action has been validated, the action will be submitted to the Acting stage for execution. The revised action is still probably erroneous, causing run-time errors in the software. In this case, the state machine will return to the Revising stage to prompt the LLM to re-plan according to the error information until the execution is finished. The entire process is repeated until the LLM outputs "Done!".

## C.2  Prompting LLMs to Generate Task Solutions

The initial prompt used to query the LLM to generate step-by-step solutions is shown in Tab. I. The prompt is formatted as a multi-turn dialog according to the usage of OpenAI ChatGPT API, with each turn comprising a content field and a role field. The prompt consists of an overall task goal, a list of atomic actions, requirements, an exemplar, and finally a task to be solved.

## D  Additional Ablation Studies

## D.1  Qualitative Experiment of Missing Atomic Actions

We deliberately remove several atomic actions to test the robustness of our method. Specifically, we remove SetConditionalFormat and test SheetCopilot on a task involving conditional formatting. We also remove another important atomic action - CreatePivotTable - to see whether SheetCopilot is able to adopt other feasible solutions to analyze data. We use GPT-3.5-Turbo as the backend of our SheetCopilot. Fig. F present surprising results: Example 1 shows that SheetCopilot uses a filter to retain the rows fulfilling the task requirement, set the formats of these rows, and finally cancel the

---

[3]The model used for embedding is OpenAI text-embedding-ada-002.
[4]The model used for embedding is the same as above.

Table I: The initial prompt used to query LLMs to generate step-by-step solutions. The prompt is formatted as a multi-turn dialogue between a user and an agent. Note that the final turn of this prompt is an example task instruction and can be replaced with other task instructions to generate solutions corresponding to the task. For clarity, redundant contents are left out.

{content:You are a spreadsheet agent who can find proper action APIs from the API document based on the language instructions.
Here is the API document:
Write # Args: (range: str, value: str) Usage: Write value into a range. The string in value also can be excel formulas.
CopyPaste # Args: (source: str, destination: str) Usage: Copy the source range and paste into the destination range.
CutPaste # Args: (source: str, destination: str) Usage: Cut the source range and paste into the destination range.
SetHyperlink # Args: (source: str, url: str) Usage: Set hyperlink for the source range.
RemoveHyperlink # Args: (source: str) Usage: Remove hyperlink for the source range.
AutoFill # Args: (source: str, destination: str) Usage: Auto fill the destination range with the source range.
Sort # Args: (source: str, key1: str, order: str='asc', orientation: str='column') Usage: Sort the source range by key1.
Filter # Args: (source: str, fieldIndex: int, criteria: str) Usage: Filter the source range based on fieldIndex by criteria.
DeleteFilter # Args: () Usage: Delete all filters.
MoveRow # Args: (source: int, destination: int) Usage: Move the source row to the destination row.
MoveColumn # Args: (source: int, destination: int) Usage: Move the source column to the destination column.
RemoveDuplicate # Args: (source: str, key: int) Usage: Remove duplicate values in the source range based on the key.
SetFormat # Args: (source: str, font: str = None, fontSize: float = None, color: str = None, fillColor: int = None, bold: bool = None, italic: bool = None, underline: bool = None, horizontalAlignment: str = None) Usage: Set format for the source range. If you want to set data type, please use 'SetDataType' API.
SetDataType # Args: (source: str, dataType: str) Usage: Set data type for the source range.
SetCellMerge # Args: (source: str, merge: bool) Usage: Toggle cell merge for the source range.
Delete # Args: (source: str, region: str) Usage: Deletes a cell or range of cells.
Clear # Args: (source: str) Usage: Clear the content and the formatting of a Range.
AutoFit # Args: (source: str) Usage: Auto fit the width and height of all cells in the range.
ResizeRowColumn # Args: (source: str, width: float = None, height: float = None) Usage: Resize the width and height of all cells in the range.
SetConditionalFormat # Args: (source: str, formular: str, bold: nool = None, color: str = None, fillColor: str = None, italic: bool = None, underline: bool = None) Usage: Set conditional format for the source range.
... (Leave out for clarity)
FreezePanes # Args: (source: str) Usage: Freeze panes for current window.
CreateChart # Args: (source: str, destSheet: str, chartType: str, chartName: str, XField: int = None, YField: List[int] = []) Usage: Create a chart based on the data from the source range. Please note that if you use data from a pivot table to create a chart, use the API 'CreateChartFromPivotTable' instead.
... (Leave out for clarity)
Requirements:
1. The user will show you the headers and row numbers of sheets for you reference.
2. Please provide step-by-step solutions with explanations.
3. You can only be allowed to use the action APIs listed above. You cannot use any other APIs. Do not generate any new action APIs.
4. It should only return one step at a time and each step should only contain one action API.
5. Please add @ both before and after each function call to indicate that the content between the two @ characters is one function call, like @Function1()@, Function2()@.
6. The address of a range must contain the sheet name, due to that there may be multiple sheets in a workbook.
7. You can only call the action APIs and directly use the values in the sheet. You cannot use any other variables.,
role:system}
{content:Instruction: In column D, calculate the profit for each week. Then format the numbers with Accounting Number Format.
Sheet state: Sheet "Sheet1" (active) has 3 columns (Headers are A: "Week", B: "Sales", C: "COGS") and 11 rows (the row 1 is the header row).
Please provide the first step.,
role:user}
{content:Step 1. Create a new column D.
Action API: @Write(range="Sheet1!D1", value="Profit")@,
role:assistant}
{content:Sheet state: Sheet "Sheet1" has 4 columns (Headers are A: "Week", B: "Sales", C: "COGS", D: "Profit") and 11 rows (the row 1 is the header row).
If task is not finished, please provide the next step, otherwise, please type "Done!".,
role:user}
{content:Step 2. Profit is sales minus COGS.
Action API: @Write(range="Sheet1!D2", value="=B2-C2")@,
role:assistant}
{content:Sheet state: Sheet "Sheet1" has 4 columns (Headers are A: "Week", B: "Sales", C: "COGS", D: "Profit") and 11 rows (the row 1 is the header row).
If task is not finished, please provide the next step, otherwise, please type "Done!".,
role:user}
{content:Step 3. Fill other rows.
Action API: @AutoFill(source="Sheet1!D2", destination="D2:D11")@,
role:assistant}
{content:Sheet state: Sheet "Sheet1" has 4 columns (Headers are A: "Week", B: "Sales", C: "COGS", D: "Profit") and 11 rows (the row 1 is the header row).
If task is not finished, please provide the next step, otherwise, please type "Done!".,
role:user}
{content:Step 4. Change the format of the results as these are accounting values.
Action API: @SetDataType(range="Sheet1!D2:D11", dataType="currency")@,
role:assistant}
{content:Sheet state: Sheet "Sheet1" has 4 columns (Headers are A: "Week", B: "Sales", C: "COGS", D: "Profit") and 11 rows (the row 1 is the header row).
If task is not finished, please provide the next step, otherwise, please type "Done!".,
role:user}
{content:Done!,
role:assistant}

(The following turn shows the context, instruction, and sheet state of the task to be solved)
{content:My workbook has two tables. Sheet "Sheet1" records the sales of a boomerang company. Sheet "Retail Price" lists the retail prices for all products.
Instruction: I want to calculate the revenue for each transaction in the sales table considering corresponding retail price and discount. Please help me do this in a new column with header "Revenue". Then create a pivot table at A1 in a new sheet to show the counts of the websites on which boomerangs were sold. Finally, plot a bar chart in this new sheet for the pivot table with chart title "Sales frequency by website" and without legends.
Sheet state: Sheet "Sheet1" has 6 columns (Headers are A: "Date Time", B: "Web Site", C: "Product", D: "Type", E: "Quantity", F: "Discount") and 36 rows (1 header row and 35 data rows). Sheet "Retail Price" has 2 columns (Headers are A: "Product", B: "Retail Price") and 23 rows (1 header row and 22 data rows).
Please provide the first step.,
role:user}

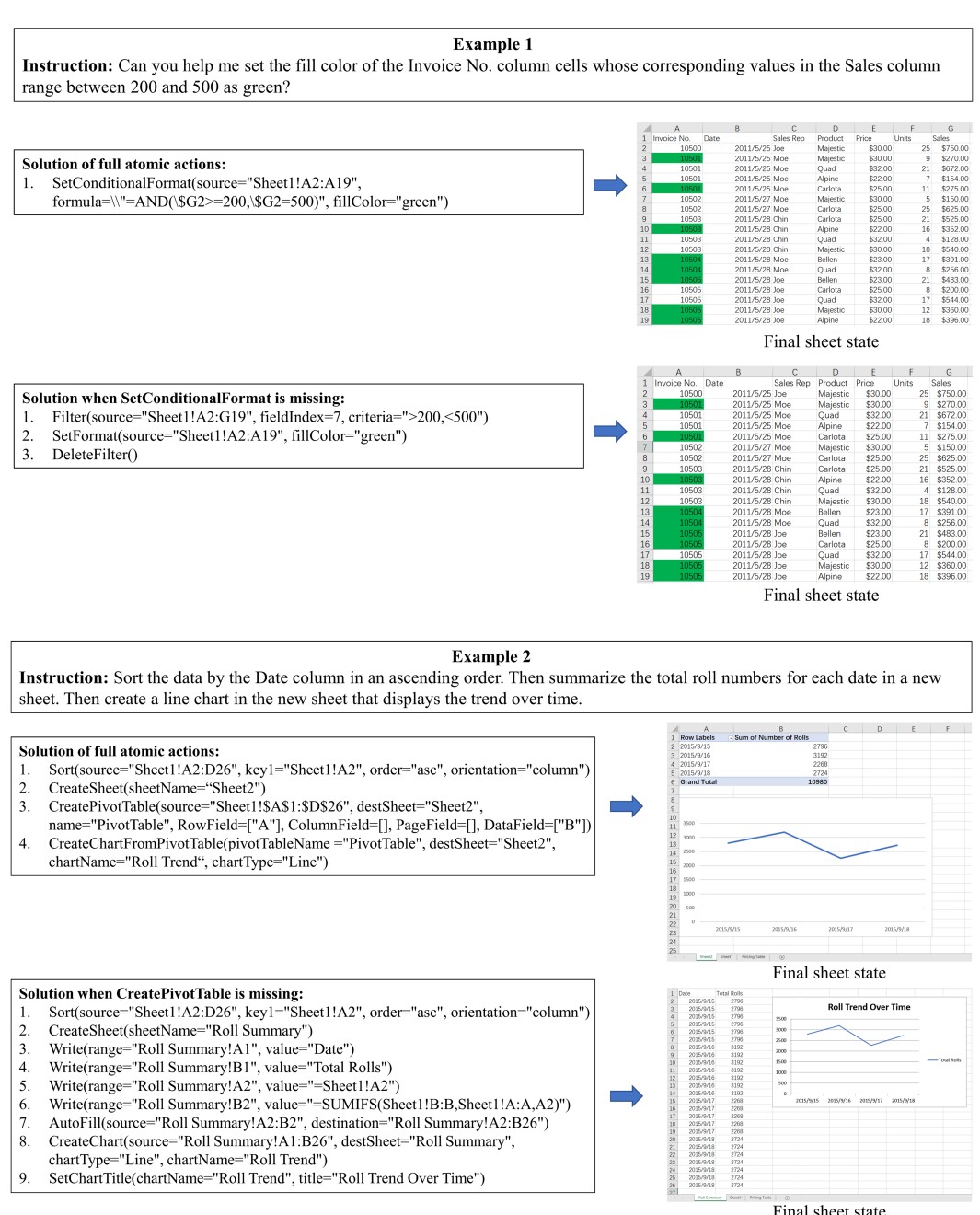

Figure F: Two example solutions generated with missing atomic actions. For each example, the instruction (top), solutions generated with full/missing atomic actions (left), and the final sheet states after implementing the two solutions (right) are displayed. The two examples show that SheetCopilot demonstrates strong robustness by using a combination of other atomic actions to attain the same goal which can be more efficiently attained using the missing atomic action.

filter to restore the data. Example 2 shows that SheetCopilot cleverly utilizes the SUMIF function to substitute the pivot table. Despite redundant calculation, it obtains functionally correct results.

## D.2 Detailed Analysis of Atomic Action at Different Granularity

We provide further details about the ablation studies on the action granularity (Sec. 5.6 in the main paper). In our design, the low-level implementations of the atomic action used by SheetCopilot are consistent with the atomic actions used by Excel in terms of behavior, which means that our atomic actions possess a normal granularity and are familiar to average users. However, one may argue that a high-level atomic action that combines several actions used by our SheetCopilot may bring better performances as using high-level actions possibly improves efficiency.

To investigate this interesting problem, we combine several chart-related atomic actions into one high-level chart manipulation action and conduct an ablation study on the tasks involving charts. Specifically, we incorporate the actions that set chart properties into the CreateChart action and remove the original separate actions, expecting the SheetCopilot to set all required properties on the creation of the chart. Additionally, the usages of these actions are merged and a new example of showing how to use this integrated CreateChart action is also added (See the documents of the separate actions and integrated action in Tab. J). Additionally, in another experiment, we split SetFormat which originally contains multiple arguments related to setting various format properties to see the impact of using finer-grained format-setting actions (See the documents of the SetFormat action and finer-grained format-setting actions in Tab. K). The SetConditionalFormat action possesses a function overlapped with that of SetFormat, thus interfering with the experiment. Therefore, we remove the SetConditionalFormat action when conducting this experiment.

We conduct this ablation study with GPT-3.5-Turbo as the SheetCopilot backend on the chart-related tasks (43 out of 221) and format-related tasks (41 out of 221). The results in Tab. 4 show that using a high-level CreateChart action to handle chart creation and chart property setting simultaneously obtains slightly higher efficiency (lower A50 and A90). However, it is surprising that this method encounters significantly inferior functional correctness, with Exec@1 and Pass@1 decreasing by 12.6% and 6.1 %, respectively. Likewise, splitting the original SetFormat action witnesses considerable performance gains in Exec@1 and Pass@1.

After analyzing the chart-related results, we found that using an integrated CreateChart encounters lower functional correctness since its document is so complex that SheetCopilot struggles to understand the action usage, thus being less able to correctly determine all action arguments as required. In addition, the document of this integrated CreateChart action is extremely lengthy, causing the query to frequently exceed the GPT-3.5-Turbo token limit. These results suggest that it is more desirable to use finer-grained atomic actions instead of integrated high-level actions in terms of functional correctness. After analyzing the format-related results, we observed that using the original SetFormat action tends to result in argument hallucination. For example, the LLM frequently adds a non-existing argument such as "criteria=...", trying to use SetFormat in a way similar to the removed SetConditonalFormat. The LLM outputs such erroneous action repeatedly even if the error feedback is provided, finally exceeding the token limit and causing an execution error. In contrast, after splitting the SetFormat action, the LLM is able to easily understand the simple finer-grained actions, thereby encountering fewer hallucination cases.

## E    SheetCopilot Manipulation Example

To better illustrate the strengths of our SheetCopilot, an example of using SheetCopilot (GPT-3.5) to produce solutions to users' requests is shown in Fig. G. SheetCopilot is asked to complete the sales data, analyze the results, and visualize the trend. SheetCopilot accurately understands the intention of the user's instruction and correctly revises its solution utilizing the external document of the chosen atomic action and the error feedback provided by the evaluation environment.

## F    Analysis of failure cases

To obtain a clearer view of the distinction between the LLMs compared in Sec. 5.2, we perform an in-depth analysis of the failure cases for each LLM. The predefined failure types are as follows:

Table J: The documents of the separate chart-related atomic actions (left column) and the integrated CreateChart action (right column). The arguments and usages of each separate action are merged and a new example of using the integrated CreateChart is provided in the document of the integrated CreateChart action. For clarity, the redundant contents are substituted with ellipsis.

| The documents of the separate chart-related actions | The document of the integrated CreateChart action |
|---|---|
| CreateChart:
args: "(source: str, destSheet: str, chartType: str, chartName: str, XField: int = None, YField: List[int] = [])"
args explanation:
source (string): The range which contains the data used to create the chart.
destSheet (string): The name of the sheet where the chart will be located.
chartType (string): The type of chart. It can be 'Area', 'AreaStacked', 'BarClustered', 'BarOfPie', 'BarStacked', 'Bubble', 'ColumnClustered', 'ColumnStacked', 'Line', 'LineMarkers', 'LineMarkersStacked', 'LineStacked', 'Pie', 'XYScatter', 'XYScatterLines', 'XYScatterLinesNoMarkers', 'XYScatterSmooth', 'XYScatterSmoothNoMarkers', '3DPie'.
chartName (string): The name for the chart to be created.
XField (int): The index of the column which contains the X values, starting from 1. If XField is None, the first column will be used.
YField (List[int]): The indices of the columns which contain the Y values, starting from 1. If YField is [], all columns except the first column will be used.
usage: Create a chart based on the data from the source range. Please note that if you use data from a pivot table to create a chart, use the API 'CreateChartFromPivotTable' instead.
example:
# Example 1: Create a chart in Sheet2 based on the data from A1 to B10 in Sheet1 and set the chart name to 'Chart1'.
CreateChart(source='Sheet1!A1:B10', destSheet='Sheet2', chartType='ColumnClustered', chartName='Chart1')
# After implementing this action, a chart named 'Chart1' will be created in Sheet2 based on the data from A1 to B10 in Sheet1.
...

SetChartTitle:
args: "(chartName: str, title: str, fontSize: float = None, bold: bool = None, color: str = None)"
args explanation: ...
example:

SetChartHasAxis:
args: "(chartName: str, axis: str, hasAxis: bool)"
args explanation: ...
example:

SetChartAxis:
args: "(chartName: str, axis: str, title: str = None, labelOrientation: str = None, maxValue: float = None, miniValue: float = None)"
args explanation: ...
example:

SetChartHasLegend:
args: "(chartName: str, hasLegend: bool)"
args explanation: ...
example:

SetChartLegend:
args: "(chartName: str, position: str = None, fontSize: str = None, seriesNamelist = [])"
args explanation: ...
example:

SetChartType:
args: "(chartName: str, chartType: str)"
args explanation: ...
example:

AddDataLabels:
args: "(chartName: str)"
args explanation: ...
example:

AddChartErrorBars:
args: "(chartName: str)"
args explanation: ...
example:

SetChartMarker:
args: "(chartName: str, style: List[str] = None, size: float = None)"
args explanation: ...
usage: Set marker for the chart.
example: | CreateChart:
args: "(source: str, destSheet: str, chartType: str, chartName: str, XField: int = None, YField: List[int] = [], title: str = None, titleSize: float = None, titleBold: bool = None, titleColor: str = None, hasLegend: bool = None, legendPosition: str = None, legendSize: float = None, legendNames: list = [], hasErrorBars: bool = None, hasDataLabels: bool = None, markerStyle: List[str] = None, makerSize: float = None, trendlineType: List[str] = None, trendlineDisplayEquation: bool = None, trendlineDisplayRSquared: bool = None, hasXAxis: bool = None, XAxisTitle: str = None, hasYAxis: bool = None, YAxisTitle: str = None)"
args explanation: ...
usage: Create a chart based on the data from the source range and also set all properties at creation time. Note that it is not allowed to set the properties for an existing Chart. Please note that if you use data from a pivot table to create a chart, use the API 'CreateChartFromPivotTable' instead.
example:
# Example 1: Create a chart in Sheet2 based on the data from A1 to B10 in Sheet1 and set the chart name to 'Chart1'.
CreateChart(source='Sheet1!A1:B10', destSheet='Sheet2', chartType='ColumnClustered', chartName='Chart1')
# After implementing this action, a chart named 'Chart1' will be created in Sheet2 based on the data from A1 to B10 in Sheet1.
# Example 2: Create a chart based on the data from A1 to B10 in Sheet1 and set the chart title to 'Chart1 Title'.
CreateChart(source='Sheet1!A1:B10', destSheet='Sheet1', chartType='ColumnClustered', chartName='Chart1', title='Chart1 Title')
# After implementing this action, a chart named 'Chart1' will be created in Sheet1 based on the data from A1 to B10 and the chart title will be set to 'Chart1 Title'.
# Example 3: Create a chart named 'Chart1' and set title, marker, X Y-axis titles, legend, and trendline.
CreateChart(source='Sheet1!A1:C10', destSheet='Sheet1', chartType='ColumnClustered', chartName='Chart1', XField=1, YField=[2,3], title='Chart1 Title', hasLegend=True, legendPosition='bottom', markerStyle=['circle','triangle'], trendlineType=['polynomial'], trendlineDisplayEquation=True, trendlineDisplayRSquared=True, hasXAxis=True, XAxisTitle='X-axis', hasYAxis=True, YAxisTitle='Y-axis')
# After implementing this action, a chart named 'Chart1' will be created in Sheet1 based on the data from A1 to C10 and the chart title will be set to 'Chart1 Title'. The first column will be used as X values and the second and third columns will be used as Y values. The legend will be displayed at the bottom. The first series will have circle marker and the second series will have triangle marker. A polynomial trendline will be added to the chart and the equation and R squared will be displayed. The X-axis title will be set to 'X-axis' and the Y-axis title will be set to 'Y-axis'. |

Table K: The documents of the finer-grained format-setting actions (left column) and the original SetFormat action (right column). The argument and usage of the original SetFormat are both divided as the arguments and usages of the finer-grained actions. For clarity, the redundant contents are substituted with ellipsis.

| The documents of the finer-grained format-setting actions | The document of the original SetFormat action |
| --- | --- |
| SetFont:
args: "(source: str, font: str)"
args explanation:
source (string): The range to set font.
font (string): The font to set.
usage: Set font for the source range.
example:
# Example 1: Set font for the range (A1:B6) to 'Arial'.
SetFont(source="Sheet1!A1:B6", font="Arial")
# After implementing this action, the range (A1:B6) will be set to 'Arial' font.

SetFontSize:
args: "(source: str, fontSize: float)"
args explanation:
...
usage: Set font size for the source range.
example:
# Example 1: Set font size for the range (A1:B6) to 20.
SetFontSize(source="Sheet1!A1:B6", fontSize=20)
# After implementing this action, the range (A1:B6) will be set to 20 font size.

SetFontColor:
args: "(source: str, color: str)"
args explanation:
...
usage: Set font color for the source range.
example:
# Example 1: Set font color for the range (A1:B6) to 'red'.
SetFontColor(source="Sheet1!A1:B6", color="red")
# After implementing this action, the range (A1:B6) will be set to 'red' font color.
SetFillColor:
args: "(source: str, fillColor: str)"
args explanation:
...
usage: Set fill color for the source range.
example:
# Example 1: Set fill color for the range (A1:B6) to 'red'.
SetFillColor(source="Sheet1!A1:B6", fillColor="red")
# After implementing this action, the range (A1:B6) will be set to 'red' fill color.
SetBold:
args: "(source: str, bold: bool)"
args explanation:
...
example:
# Example 1: Set bold for the range (A1:B6).
SetBold(source="Sheet1!A1:B6", bold=True)
# After implementing this action, the range (A1:B6) will be set to bold.
SetItalic:
args: "(source: str, italic: bool)"
args explanation:
...
example:
# Example 1: Set italic for the range (A1:B6).
SetItalic(source="Sheet1!A1:B6", italic=True)
# After implementing this action, the range (A1:B6) will be set to italic.

SetUnderline:
args: "(source: str, underline: bool)"
args explanation:
...
usage: Set underline for the source range.
example:
# Example 1: Set underline for the range (A1:B6).
SetUnderline(source="Sheet1!A1:B6", underline=True)
# After implementing this action, the range (A1:B6) will be set to underline.

SetHorizontalAlignment:
args: "(source: str, horizontalAlignment: str)"
args explanation:
source (string): The range to set horizontal alignment.
horizontalAlignment (string): The horizontal alignment to set. It can be 'left', 'center', 'right'.
usage: Set horizontal alignment for the source range.
example:
# Example 1: Set horizontal alignment for the range (A1:B6) to 'left'.
SetHorizontalAlignment(source="Sheet1!A1:B6", horizontalAlignment="left")
# After implementing this action, the range (A1:B6) will be set to 'left' horizontal alignment. | SetFormat:
args: "(source: str, font: str = None, fontSize: float = None, color: str = None, fillColor: int = None, bold: bool = None, talic: bool = None, underline: bool = None, horizontalAlignment: str = None)"
args explanation:
source (string): The range to set format.
font (string): The font to set.
fontSize (float): The font size to set.
color (string): The color to set. It can be 'black', 'white', 'red', 'green', 'blue', 'yellow', 'magenta', 'cyan', 'dark_red', 'dark_green'.
fillColor (string): The fill color to set. It can be 'black', 'white', 'red', 'green', 'blue', 'yellow', 'magenta', 'cyan', 'dark_red', 'dark_green'.
bold (bool): Whether to set bold. True means bold, False means not bold.
talic (bool): Whether to set talic. True means talic, False means not talic.
underline (bool): Whether to set underline. True means underline, False means not underline.
horizontalAlignment (string): The horizontal alignment to set. It can be 'left', 'center', 'right'.
usage: Set format for the source range. If you want to set data type, please use 'SetDataType' API.
example:
# Example 1: Write bold text "Summer sales ($)" with blue fill color and white text color in A1.
Write("Sheet1!A1", "Summer sales ($)")
SetFormat("Sheet1!A1", bold=True, fillColor="blue", color="white")
# After implementing this action, the cell A1 will contain bold text "Summer sales ($)" with blue fill color and white text color.
# Example 2: Adjust column C to Arial font with underline.
SetFormat("Sheet1!C:C", font="Arial", underline=True)
# After implementing this action, the column C will be adjusted to Arial font with underline. |

Figure G: SheetCopilot example: Handling sales data. The left column shows that SheetCopilot generates a step-by-step solution according to the sheet state feedback and correctly revises its mistakes using the external atomic action document as well as the error feedback. The incorrect arguments are marked with red rectangles. The right column demonstrates the state changes of the evaluation sheet corresponding to each step on the left. For illustration clarity, only brief documents are displayed.

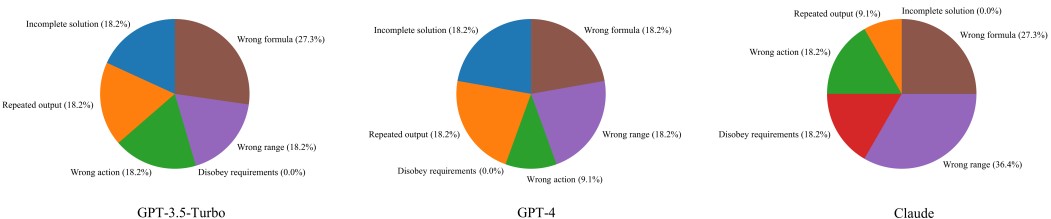

GPT-3.5-Turbo

GPT-4

Claude

Figure H: Comparing the proportions of different failure cases of the three LLMs. The evaluation dataset is the 10% subset. The charts show that GPT-3.5-Turbo and GPT-4 share similar failure patterns while Claude exhibit a different one.

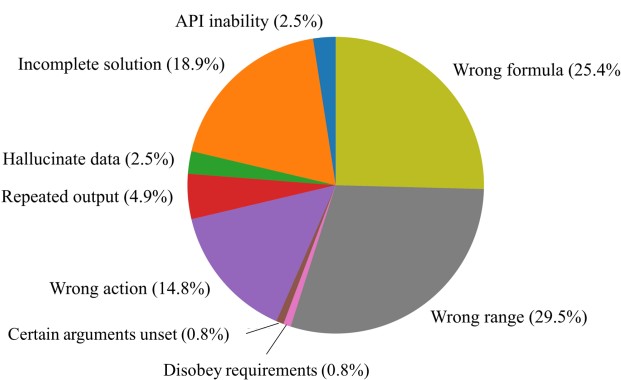

Figure I: The proportions of different failure cases when evaluating GPT-3.5-Turbo on the full coreset.

- **Wrong action** Use wrong atomic actions to attain a specific goal. For instance, hide columns using a filter (a filter can only be used to hide rows) or perform meaningless actions to destroy the data.
- **Wrong range** Select the wrong ranges to manipulate. For instance, fail to use absolute references when required or select incomplete ranges as chart data sources.
- **Certain arguments unset** Perform the atomic actions required by the tasks but fail to set all required arguments. For instance, set the cell background color but forget to set the cell font color.
- **Repeated output** Output the same atomic action repeatedly until exceeding the token limit.
- **Hallucinate data** Hallucinate data to manipulate while ignoring the real data.
- **Incomplete solution** Terminate a solution without fulfilling all task requirements.
- **Disobey requirements** Perform an atomic action in a way that deviates from the task requirements. For example, insert a hyperlink different from the one specified in the task instruction.
- **API inability** Failure caused by the limitations of the low-level implementations of the atomic actions. For instance, the low-level implementation of conditional formatting does not support all types of formulas (e.g. array formulas or formulas available only in a specific software version).

The categorized failure cases of the three LLMs are shown in Fig. H. From an overall viewpoint, we can obtain two **interesting findings**: 1) GPT-3.5-Turbo and GPT-4 share similar patterns of the failure case proportions while Claude demonstrates a largely different failure pattern. This finding suggests that the two GPT models possess almost identical alignment in terms of spreadsheet manipulation and that a gap exists between the alignment of Claude and those of the GPT models. 2) Jointly analyzing the failure cases and the performances in Tab. 1, we can see that although the prompt used by SheetCopilot (See the prompt in Tab. I) is tuned for GPT-3.5-Turbo, Claude exhibits competitive performances compared with GPT-3.5-Turbo.

Inspecting individual cases, we found that GPT-3.5-Turbo tends to use wrong formulas, showing that its ability to predict formulas is weak. GPT-4 is prone to incomplete solutions due to the limited context window. In contrast, Claude encounters no incomplete solutions and less frequent repeated output but tends to use wrong formulas and wrong ranges. One common failure case among the three LLMs is that they often use a combination of UNIQUE and SUMIF/AVERAGEIF/COUNTIF to replace pivot tables. This combination is technically correct but these three models forget to apply absolute reference when selecting manipulated ranges used by auto-filling, leading to incorrect calculation results.

We also check the failure cases made by GPT-3.5-Turbo on the full core set (See the experimental results in Tab. 1) and illustrate the percentages of these cases in Fig. I. The pie chart shows that GPT-3.5-Turbo performs poorly at predicting formulas and determining appropriately selected ranges. These two types of failure occupy over 50% of all cases. Inspecting the wrong formula cases, we

found that GPT-3.5-Turbo often fails to use absolute references when necessary, uses self-reference, and uses wrong criteria for a filter or conditional formatting. Inspecting the wrong range cases, we found that GPT-3.5-Turbo tends to use wrong auto-filling ranges, use wrong chart data sources, and leave out the headers when copy-pasting columns. Another two prominent types of failure are Wrong action and Incomplete solution, which occupy 33.7% in total. After inspecting the wrong action cases, we found that GPT-3.5-Turbo misunderstands the usage of the atomic actions it incorrectly uses. For example, GPT-3.5-Turbo inserts an unnecessary column to fill in data, ignoring the existing column where the data should be written; GPT-3.5-Turbo also uses a filter to hide columns, which violates the actual usage of a filter. After inspecting the incomplete solutions, we found that GPT-3.5-Turbo fails to fulfill all task requirements in these cases although other steps are correct (e.g. it forgets to set the chart title or to autofill after filling in the first row). In the cases where repeated output occurs, GPT-3.5-Turbo generates the same action repeatedly as it fails to revise its plan according to the given error feedback. GPT-3.5-Turbo occasionally misses setting required arguments or disobeys the task requirements to set incorrect arguments which both make up a small fraction of the failure cases. A small proportion of the failure cases result from the implementation limitations of several atomic actions, which can be fixed by upgrading these implementations.

## G   Demo

Video demos for our SheetCopilot agent can be found on our project website (https://sheetcopilot.github.io).

## H   Limitations

Our SheetCopilot possesses the following limitations:

- We have not optimized the number of tokens used to solve spreadsheet tasks. This means that SheetCopilot consumes a large number of tokens and is only capable of solving short-horizon tasks. Future work is needed to focus on designing efficient prompting methods that save tokens or using LLMs with a larger context window.

- State feedback only consists of the basic information of cells, ignoring the state of charts, pivot tables, filters, and frozen panes. SheetCopilot probably creates charts and pivot tables repeatedly as it is unable to observe the outcome of creating these elements.

- Our dataset is manually curated and is not fully fledged yet. More labor work is required to generate more ground truth as numerous correct solutions to each task probably exist. Moreover, the more solutions are collected to conduct evaluation, the more accurate and stable performances we can obtain.

- Our evaluation environment does not support all Excel built-in features, which restricts the task categories our SheetCopilot is able to handle. More effort is needed to continuously upgrade the evaluation system to accommodate more diverse tasks.

## I   Broader Impact

Our proposed SheetCopilot possesses the potential to significantly simplify the process of spreadsheet control, which positively impacts productivity and efficiency. By enabling users to interact with spreadsheets using natural language, SheetCopilot reduces the time and effort required to perform a wide range of tasks, such as simple data entry and complex data analysis.

However, it is imperative to consider potential negative impacts. One concern is that the increased ease of spreadsheet control possibly leads to an over-reliance on automation, potentially reducing the need for human expertise in certain areas, which may cause unemployment in society. Additionally, there is a risk that the use of SheetCopilot exacerbates existing inequalities in the workforce as those who lack access to SheetCopilot or who are not proficient in using it may be at a disadvantage.

Another potential risk of using SheetCopilot is that a wrong operation is likely to face undesirable consequences when controlling sensitive data forms, such as tax forms or financial statements. This risk highlights the importance of ensuring the robustness and accuracy of the agent's actions, as well as the need for appropriate safeguards and oversight to minimize the risk of errors.

Additionally, SheetCopilot relies on the capabilities of language models, which are known to potentially inherit biases present in the training data. As a result, SheetCopilot may inadvertently exacerbate existing biases when interacting with spreadsheets and processing data. It is crucial to acknowledge the potential for inherited bias from LLMs in the development and deployment of SheetCopilot and to take proactive steps to mitigate any negative impacts on fairness and equity.

