# OpenReview forum: "SheetCopilot: Bringing Software Productivity to the Next Level through Large Language Models"
_NeurIPS.cc/2023/Conference — NeurIPS 2023 poster_

### Official Review · Reviewer_HU2o · 2023-07-04

**Soundness:** 3 good
**Presentation:** 3 good
**Contribution:** 2 fair
**Rating:** 6
**Confidence:** 4

**Summary:**

Authors create SheetCopilot that takes input as a natural language task and then controls spreadsheet to perform the task. To provide an interface between natural language and the spreadsheet, author use a set of atomic actions that abstract spreadsheet functions. SheepCopilot uses state machine to create spreadsheet actions, modify them, and apply them to the spreadsheet.
Authors also present a dataset to evaluate their tool. SheetCopilot outperforms baseline approaches

**Strengths:**

- Authors propose an approach to interface LLMs with spreadsheets potentially automating a large number of repetitive tasks
- Intermediate language of atomic actions provides a good output structure for LLM
- State machine based processing of tasks improves the results
- Atomic action name substitution with synonyms that are far away from the official names in an embedding space is interesting approach (but see weaknesses)
- Test dataset is useful for evaluating SheetCopilot and similar systems
- Ablation studies show clearly the contributions of different parts of the system
- Positive results compared to VBA generation baseline
- Paper is well written

**Weaknesses:**

- Paper contributions are moderate:
- Intermediate language similar to atomic actions is a known approach in software engineering, similar to macro or scripting languages.
- Atomic actions have to be implemented in spreadsheet software which requires more effort to adopt this approach compared to existing interfaces to spreadsheets. (This is not a major weakness though, especially if current interfaces to spreadsheets are not well suited or universal).
- Atomic action name substitution with synonyms seems like a hacky approach, a bit similar to security-through-obscurity. It may have negative consequences (e.g. making it harder for LLM to connect tasks and actions) and it is not clear if this is should be used or relied upon in general.

I have read the author’s rebuttal.
The rebuttal partially addressed my concerns about atomic actions.

**Questions:**

Why do certain tasks fail? Are the failures in certain categories due to the tasks, some limitations of LLMs, selection of atomic actions, spreadsheet functionality, too simple state machine? (Also see limitations comments).

I have read the author’s rebuttal.
The rebuttal addressed my concerns about understanding of why certain tasks fail by providing failure analysis.

**Limitations:**

It would be good to have a more in-depth understanding why certain tasks fail and where are the boundaries of LLM spreadsheet interactions. It is good that authors classified tasks into six categories and observed different failure rates across them. However, it would be good to go further and understand the failures in more depth (also see questions).

I have read the author’s rebuttal.
The rebuttal addressed my concerns about understanding of why certain tasks fail by providing failure analysis.

---

> ### Author Rebuttal · Authors · 2023-08-10
>
> Thank you for a thorough review that will help us improve the work. Please see below for answers to your questions.
>
> **W1: Paper contributions are moderate.**
>
> We believe our work contributes to the field of tool-augmented LLM, for three primary reasons:
>
> - We propose a novel framework enabling model-software interaction. The closed-loop control (Section 4.3) boosts success from 67.8% to 87.3%  (row 2 vs. row 7 in Table 2). Our framework also substantially outperforms a VBA baseline (Table 1). This facilitates applying language models to software automation.
>
> - Our dataset contains a diverse range of daily spreadsheet tasks (e.g., formulas, charts, and pivot table tasks). We believe this is a meaningful contribution because our work contributes a procedure for collecting software task datasets as well as a new testbed for comparing the planning capabilities of language models. To our knowledge, no existing dataset offers equally broad spreadsheet tasks, making ours useful for research on LLM-based agents.
>
> - We thoroughly compare leading LLMs (Table 1 in the main paper and failure analysis in the Supplementary), elucidating the strengths and weaknesses of these well-known models from an aspect of task planning.
>
> In summary, the proposed SheetCopilot, dataset, and experiments provide novel contributions. We would appreciate any suggestions to further communicate the value of our research to the readers.
>
> **W2: Intermediate language similar to atomic actions is a known approach**
>
> We agree that abstraction is standard in software engineering. However, to the best of our knowledge, no prior work has designed a unified abstraction for diverse spreadsheet platforms. Our atomic actions pioneer such an interface between LLMs and spreadsheets, analogous to an intermediate language abstracting assembly languages.
>
> Our atomic actions offer two key advantages:
>
> - They model spreadsheet functionality as virtual APIs (Section 4.2), enabling cross-platform control, as evidenced by Excel and Google Sheets compatibility (Fig. (b) in the response PDF). In contrast, macro languages like VBA typically target specific platforms.
>
> - They can be used by LLMs to generate human- and machine-readable solutions (Figures F and G in the supplementary).
>
> In summary, our atomic actions constitute a novel, unified spreadsheet abstraction.
>
> **W3: Atomic actions have to be implemented in spreadsheet software**
>
> It is right that current spreadsheet interfaces lack universality. This forces LLMs to relearn new rules when controlling new software, complicating prompting and debugging.
>
> To address this, we propose software-agnostic atomic actions (Section 4.2) as a high-level abstraction. This enables cross-platform compatibility, as evidenced by SheetCopilot for Excel and GoogleSheets (Fig. (b)).
>
> In summary, the platform-agnostic nature of our atomic actions is a strength rather than a weakness.
>
> **W4: Atomic action name substitution seems hacky**
>
> We would like to clarify that atomic action name substitution is a component of our method, instead of a standalone approach.
>
> The goal is mitigating a well-known LLM issue - hallucination - where models generate confident response that is not justified by their training data or that contradicts input prompts [1][2]. For example, we observed GPT-3.5 hallucinating a non-existing argument such as “criteria=” and using SetFormat in a way similar to SetConditonalFormat).
>
> The cause is that GPT-3.5 confuses the atomic action knowledge with its internal Excel knowledge. GPT-3.5, like other LLMs, generates responses based on patterns it learned during training [3]. It has no access to real-time data (e.g., the usage of atomic actions) and cannot absorb new concepts after its training is complete.
>
> Therefore, if language patterns of the atomic actions are similar to those it has learned about Excel operations, GPT-3.5 will probably use the actions in a way like Excel operations. Figure I in Supplementary reveals such hallucination causes failures.
>
> To tackle this, we substitute official names with dissimilar synonyms, enabling LLMs to strictly follow the usage of the atomic actions (Section 4.4). This is supported by Table 3 showing substitution improves both Pass@1 and efficiency.
>
> In conclusion, atomic action name substitution is part of our method, playing a vital role in alleviating hallucination.
>
> **Q1: Why do certain tasks fail & Limitations: need understanding of why certain tasks fail**
>
> We appreciate your suggestion to analyze task failures.
>
> To clarify, we conducted a comprehensive failure analysis (Section F, Supplementary). We found that:
>
> - GPT-3.5 often applies incorrect formulas. For instance, it failed to use an approximate match in VLOOKUP when querying a price table.
> - GPT-4 occasionally generates incomplete solutions due to token limit. For instance, GPT-4 output correct mid-steps but the tokens had run out.
> - Claude tends to disobey requirements, like summarizing quantity rather than the requested revenue for pivot tables.
> - All three LLMs struggle with absolute references when auto-filling.
>
> Figure H compares failure patterns, showing that GPT-3.5 and GPT-4 share similar patterns while Claude exhibits a different one.
>
> Figure I delineates GPT-3.5's failure proportions over the full dataset. Failures stemming from the model itself (wrong formula/range, hallucinating, etc) occupy 73.8% in total. The failures caused by the token limit of the OpenAI API (Incomplete solutions) occupy 18.9%. The failures caused by the atomic actions (API inability) make up 2.5%. This breakdown reveals incorrect GPT-3.5 predictions as the primary failure source.
>
> We hope the detailed analysis in the supplementary can address your concern.
>
> **Ref:**
>
> [1]Survey of hallucination in natural language generation, 2023
>
> [2]Shaking the foundations: delusions in sequence models for interaction and control, 2021
>
> [3]Training language models to follow instructions with human feedback, 2022

---

> > ### Comment · Reviewer_HU2o · 2023-08-13
> > **Thanks for the comments**
> >
> > I appreciate the comments and additional discussion provided by the authors.

---

### Official Review · Reviewer_TF2z · 2023-07-05

**Soundness:** 2 fair
**Presentation:** 3 good
**Contribution:** 2 fair
**Rating:** 4
**Confidence:** 4

**Summary:**

This paper introduces an LLM based spreadsheet manipulation task system from high level human language to spreadsheet manipulation tasks. The work is in the area of tool augmented LLM systems where LLM systems are used to create a chain of actions representing a complex manipulation task in a spreasheet system. The authors create a curated dataset consisting of a list of realworld spreadsheet tasks. They also build an assessment system. They study 3 popular LLMs - GPT-4, GPT-3.5turbo and Claude. They surprisingly find that GPT3.5turbo to be the best performing and also through ablation studies find that close-control action loops improve functional correctness, bringing in external docs improve correctness and surprisingly find that using synonyms far from the actual names also improve

**Strengths:**

1. The authors take on the challenging problem of automating complex modification task in spreadsheets rather than formula repair or generation. If it works can be of big benefit to spreadsheet users.
2. The authors create  a curated dataset of tasks in a systematic way.
3. They also create a systematic evaluation system for evaluating the performance of the different popular LLMs in the context of the spreadsheet software system.

**Weaknesses:**

1. I would have liked to see a better discussion on the task colletions, the Q&A pair collection, how complex these tasks are, their categorization, why they are multicategorized, what do those "practical realworld spreadsheets" look like etc.
2. The techniques using LLMs are fairly straightforward with minor modifications like using synonyms (not sure why this works) adding documents etc. What is specific to the worksheet manipulation scenario that makes LLM easy or hard or different to work with. Lack appreciation of that.
3. Further investigation into why GPT-3.5Turbo is better than even GPT-4 would be useful as it is counterintuitive.
4. Instead of just exec@1, pass@1 softer metrics would be useful in understanding where these LLMs make mistakes and where they are good.
5. Instead of using a tag cloud for dataset evaluation something detailed metrics would have been more insightful.

**Questions:**

1. Reference to Table 1 seems off. The citation says something and the table is about something else.
2. Is the dataset for creating tasks and benchmarking available to the public to compare.
3. Did you try smaller open source LLMs and see if they can be useful or how far you can get with them.
4

**Limitations:**

1. There is not much study on the nature of the dataset curated. But the paper is just at the beginning of something interesting. While it is unlikely that one may see issues with spreadsheet data in discrimination or bias or fairness it would be good to address them.

---

> ### Author Rebuttal · Authors · 2023-08-09
>
> We thank the reviewer for the detailed review and insightful comments. Below, we address the concerns:
>
> **W1: a better discussion on the task collections, the Q&A pair collection, how complex these tasks are, their categorization, why they are multi-categorized, what do practical realworld spreadsheets look like**
>
> Section A.1 of the Supplementary Material elaborates on task collection methodology.
>
> We collected ~27k spreadsheet Q&A pairs from SuperUser, filtered irrelevant pairs by keywords, further cleaned them by prompting ChatGPT to label invalid pairs, and selected representatives through clustering, resulting in a clean Q&A pair dataset.
>
> Figure A of the Supplementary Materials visualizes task complexity, with instruction lengths ranging from 20 to 530 words. Our dataset encompasses simple and long-horizon tasks requiring up to 9 actions across sheets. The tasks are stored as a .xlsx file in the supplementary zip file for ease of viewing.
>
> Task categorization occurred concurrently with LLM-based filtering (see Section A.1). Specifically, we prompted ChatGPT to classify each Q&A pair into 7 categories (i.e., Entry and Manipulation, Management, Formatting, Charts, Pivot Tables, Formulas, and Invalid). Figure (c) in the response PDF presents an example of categorization.
>
> The tasks were multi-categorized since most of them involved spreadsheet operations belonging to multiple categories. For example, a data analysis task probably requires predicting formulas (Formulas), Auto-filling by dragging down (Entry and Manipulation) and plotting charts (Charts). If singular labeling is used, ChatGPT is prone to classification ambiguity and it is harder to evaluate the diversity and complexity of our dataset.
>
> The "practical real-world spreadsheets" was also included in the supplementary zip file. Two brief examples: (i) SummerSales: The spreadsheet records the sales of a company. (ii) GDPBreakdown: The spreadsheet catalogs economic indicators over time.
>
> **W2: The techniques using LLMs are fairly straightforward with minor modifications.**
>
> We believe the state machine-based planning pipeline of SheetCopilot is a novel mechanism for solving complex spreadsheet tasks, because
>
> - Plain text chatbots cannot directly execute spreadsheet actions, unlike SheetCopilot.
> - The closed-loop control architecture (Observing Stage, line 205) enables LLMs to accomplish long-horizon, multi-sheet tasks.
> - Our proposed revising stage (line 214) leverages software feedback to recover from errors, substantially boosting execution success from 67.8% to 92.8% (row 2 vs. row 4 in Table 2). The experimental result indicates that our design is original and powerful.
>
> Action dependency and the complexity of combining spreadsheet operations make it hard for LLMs to work with spreadsheet software. Lines 196-200 in Section 4.3 explains that LLMs must comprehend how sheet state evolves after each step to succeed. For example, LLMs struggle to manipulate filtered data without understanding the resulting layout. SheetCopilot's specialized design overcomes this obstacle.
>
> **W3: Further investigation into why GPT-3.5Turbo is better**
>
> Thanks for the suggestion. Please refer to Q2 in the global response at the top.
>
> **W4: softer metrics would be useful in understanding where these LLMs make mistakes and where they are good.**
>
> We appreciate your perspective on the importance of softer metrics in understanding the performance of the LLMs.
>
> Our study does employ multifaceted soft metrics. These include success rates across task categories and fine-grained failure analysis.
>
> Figure 3 decomposes Exec@1 and Pass@1 across six categories, revealing tasks for which each LLM excels. GPT-3.5-Turbo, GPT-4, and Claude achieve peak Pass@1 on Management, Manipulation, and Manipulation tasks, respectively.
>
> Moreover, Section F of the Supplementary presents an in-depth failure analysis by tallying the number of eight predefined failure types.
>
> Statistics reveal GPT-3.5-Turbo often utilizes incorrect formulas, GPT-4 frequently generates incomplete solutions due to token limits, and Claude tends to misapply formulas and ranges. A common mistake is that they often fail to apply absolute references when auto-filling, causing calculation errors.
>
> We hope this addresses your concern.
>
> **W5: Instead of using a tag cloud for dataset evaluation something detailed metrics would have been more insightful.**
>
> The supplementary details our dataset metrics. Figure A depicts the instruction length and atomic action distributions; Figure B illustrates category proportions and diversity within the core set; Figure C enumerates category combination frequencies. These three figures together demonstrate the richness and diversity of our dataset.
>
> **Q1: Reference to Table 1 seems off.**
>
> Regarding your comment about the reference to Table 1, we thoroughly reexamined the manuscript and table in question and believe there may be a misunderstanding.
>
> Table 1 compares LLMs and a VBA-based method. Two references exist:
>
> (i) The first (line 264) cites Table 1 to demonstrate GPT-4's strong planning capability.
>
> (ii) The second (line 306) cites Table 1 to show SheetCopilot surpassing the VBA-based method.
>
> We hope this addresses your concern. If there are further ambiguities, we would gladly address them.
>
> **Q2: Is the dataset for creating tasks and benchmarking available to the public to compare.**
>
> Yes. The dataset has been publicized on PapersWithCode anonymously and also included in the Supplementary zip file for ease of viewing.
>
> **Q3: Did you try smaller open-source LLMs**
>
> Yes. Please refer to Q1 in the global response above.
>
> **Limitations: good to address discrimination or bias or fairness issues with spreadsheet data**
>
> Thank you for providing the suggestion to improve the ethical integrity of our work. Please refer to Q5 in the global response for our explanation.
>
> _We thank the reviewer again for the insightful review and feedback._

---

### Official Review · Reviewer_XbAE · 2023-07-06

**Soundness:** 3 good
**Presentation:** 4 excellent
**Contribution:** 2 fair
**Rating:** 6
**Confidence:** 4

**Summary:**

The paper proposes a benchmark and a framework based on observe-propose-revise-act for tackling spreadsheet problems.

**Strengths:**

- The paper is very well written and the ideas are neatly presented with figures and tables. It was a pleasant read!
- The ablation studies are quite interesting. It is interesting to see the effects of external documents, usage examples, state of the spreadsheet, and error messages separately. I also liked the ablation with the synonym substitution for official atomic action names.
- The paper proposes a comprehensive framework of observe-propose-revise-act, which even though shown to work on a restricted domain of spreadsheets, can be used as inspiration for other systems, though it is not clear how scaling it up to more complex scenarios will work in practice.



**Weaknesses:**

- My main concern is over the benchmark creation process. Since the problems are picked up from a public domain site, I think there is a high possibility that these LLMs might have already seen this data during pretraining. Under this assumption, it becomes hard to understand the generalization capability of the proposed framework for unseen tasks. The authors don't discuss any deduplication efforts they performed to ensure that the performance of the shown metrics is not bloated.

- Parts of the framework have been proposed by other works as discussed in the related works. However, I would consider the framework as a whole as a novel contribution. At the same time, I believe that the tasks tacked by the paper can be considered too simplistic for these LLMs ( especially since it is shown that the LLM achieves 100% performance on some categories of these tasks). It is not clear to me how the proposed framework would scale to more complex settings like a Copilot for programming where it is difficult to retrieve the relevant state of the environment, external documentation, or usage examples. Even if there exists a way to get this information, it is unlikely that it will fit within the context length available in the input prompt.

- From my understanding, there is no timeout used and the LLM is queried multiple times until it gets the right answer. If this were the case, this is not a realistic assumption when deployed in practice. Users would not wait for long periods until they get the required task accomplished.

**Questions:**

- It would be good to discuss the performance-latency tradeoff in terms of #queries to the LLM, and the memory and compute used for inference.

- It would be good to have a user satisfaction metric as well, e.g. if the users were satisfied by the time it look for the model to accomplish the task, etc.

 - Line 112: What are LLM-based filters?

- It would be good to show the improvements brought by the framework on top of relatively smaller LLMs.

**Limitations:**

There is no limitation discussed in the main paper. A discussion about the increased computing requirements during inference and generalization to complex settings should be included.

---

> ### Author Rebuttal · Authors · 2023-08-09
>
> Thank you for the insightful review and feedback. Please see below for answers to your questions.
>
> **W1: hard to understand the generalization capability … discuss any deduplication efforts**
>
> Thank you for this constructive comment.
>
> A direct way to test generalization is to split our dataset into two parts - one containing tasks whose original questions were asked before 09/30/2021 and one after. Since the data cutoff for GPT-3.5 is 2021/09, this allows us to evaluate performance on truly unseen data. Conducting this experiment, we found SheetCopilot achieves Exec@1=89.1% and Pass@1=44.0% on the before split, and Exec@1=80.4% and Pass@1=45.7% on the after split. The similar performance across splits suggests SheetCopilot exhibits generalization capability.
>
> We conducted deduplication when collecting our dataset (see Section A.3 in the supplementary). Through the adaptation process, task instructions differ from the original SuperUser questions. Tasks also apply to new sheets rather than those mentioned on SuperUser. Moreover, ground truth solutions are step-by-step, unlike the colloquial SuperUser answers.
>
> To confirm our data can be considered unseen, we calculated the maximum similarity between each task and all SuperUser questions. Similarities ranged from 0.06 to 0.16 (avg 0.10), indicating slight semantics overlap.
>
> We also calculated ROUGE scores between our tasks and SuperUser questions, finding ROUGE-1=0.38, ROUGE-2=0.17, and ROUGE -L=0.35. The low scores further confirm a slight overlap.
>
> In summary, the dataset overlap with SuperUser is minor and SheetCopilot indeed owns generalization capability.
>
> **W2: How the proposed framework would scale to more complex settings**
>
> Our main focus is to address language-instructed spreadsheet manipulation. Actually, applying our method to coding tasks presents an interesting future research direction.
>
> As a preliminary idea, we propose treating a program's logging and error flag status as part of its environment state and using API documentation for programming languages and libraries as external documents.
>
> Regarding the potential context limitation, techniques like retrieval-augmented generation based on vector databases could help mitigate this issue when working with large-scale external documents.
>
> **W3: Users would not wait for long periods**
>
> We agree that fast response is beneficial for increasing users’ satisfaction.
>
> One cause of long waiting is deadlock due to the repeated output of LLMs (this failure occupies 4.9% as shown by Figure I in the supplementary). However, SheetCopilot's solutions in these cases still contain partially correct mid-steps. Users can employ the interactive mode (see Figure (b) in the response PDF) to abort and relaunch queries, escaping deadlocks. In contrast, correcting mistakes in code-based (VBA) methods is more difficult without easy interaction.
>
> Another cause is the long inference time. Fortunately, many methods of accelerating LLM inference, such as KV-cache [1], have been proposed. These could be applied to reduce query time when deploying SheetCopilot.
>
> We will add timeout and acceleration mechanisms in future updates to improve response speed.
>
> **Q1: The performance-latency tradeoff in terms of #queries to the LLM**
>
> We agree that latency should be optimized if we want to productize SheetCopilot.
>
> Currently, the majority of inference latency stems from our lengthy system prompt (typically 2k~3k tokens), while each generated step seldom exceeds 100 tokens.
>
> A potential optimization for reducing latency is precomputing SheetCopilot's fixed prompt once and caching it via KV-cache [1] for subsequent requests, thereby reducing the query time to the time required to generate ~100 tokens.
>
> Besides, performant open-source LLMs like llama2-7b [2] could already run at a speed of >150 tokens/s on a single 4090. In the future, this could potentially enable SheetCopilot to interact with users within hundreds of milliseconds.
>
> Overall, leveraging the fast progress in open-source LLM inference infrastructure to boost SheetCopilot user experience presents an exciting direction for future work. We will incorporate a discussion of this topic in the final revision.
>
> **Q2: good to have a user satisfaction metric**
>
> Thank you for the suggestion. While human evaluation is indispensable for human-computer interaction research, it is beyond this paper's scope, which focuses on assessing LLMs' capacity for complex software control.
>
> Nevertheless, we have implemented SheetCopilot on Google Sheets (see Figure (b) in the response PDF). We will use this platform to conduct satisfaction experiments in future work.
>
> **Q3: What are LLM-based filters?**
>
> The LLM-based filter is ChatGPT used to classify Q&A pairs (see Section A.1 in Supplementary). Specifically, we prompted ChatGPT to classify each pair. After classification, we remove ~2.4k pairs labeled as “Invalid”, thereby filtering out irrelevant pairs and obtaining a cleaner dataset.
>
> Alpaca [3] and Self-instruct [4] adopted this filtering approach and inspired our work.
>
> **Q4: the improvements on top of relatively smaller LLMs**
>
> Thanks for the advice. Please refer to Q1 in the global response above.
>
> **Limitations: There is no limitation discussed in the main paper**
>
> The limitations of SheetCopilot are enumerated in Section H of the supplementary. We also list them briefly here:
>
> - We have not optimized token usage for solving spreadsheet tasks. Future efforts should save tokens or use LLMs with a larger context window.
> - Due to the token limit, state feedback provides only essential cell information.
> - Our evaluation environment does not support all Excel built-in features (e.g. array formulae).
>
> **Ref**:
>
> [1]Efficiently scaling transformer inference, 2023.
>
> [2]Llama 2: Open Foundation and Fine-Tuned Chat Models, 2023.
>
> [3]Stanford alpaca: An instruction-following llama model, 2023.
>
> [4]Self-instruct: Aligning language model with self-generated instructions, 2022.

---

> > ### Comment · Reviewer_XbAE · 2023-08-20
> >
> > Thanks for your detailed response to my comments, especially the deduplication point. I would encourage the authors to include the discussion on performance-latency in the paper.

---

> > > ### Author Response · Authors · 2023-08-21
> > > **Response to Official Comment by Reviewer XbAE**
> > >
> > > Thank you for your encouragement! We will include a performance-latency discussion in the final version and conduct further research on this topic in the future.

---

### Official Review · Reviewer_bHxV · 2023-07-07

**Soundness:** 4 excellent
**Presentation:** 4 excellent
**Contribution:** 4 excellent
**Rating:** 7
**Confidence:** 5

**Summary:**

This paper introduces SheetCopilot, a model that aims to generate step-by-step executable command sequences for software control according to the natural language description. Besides, a benchmark dataset for evaluating software control tasks is collected. Experimental results based on the dataset are reported.

**Strengths:**

-	Generating step-by-step executable command sequences for software control according to the natural language description is a valuable problem.
-	The proposed framework SheetCopilot takes a set of atomic actions as an abstraction of spreadsheet software functionalities and contains a state-machine-based task planning framework for LLMs to interact with spreadsheets, aiming to translate high-level task description into executable command sequences, which generally makes sense and is actually novel.
-	Experimental comparison against baselines (including the state-of-the-art LLMs like GPT-3.5 & GPT-4) has shown the performance advantage of the proposed method. Besides, the ablation study indicates that the design of the proposed framework is beneficial for improving performance.
-	In addition, a comprehensive dataset containing 221 spreadsheet-related tasks is collected in this work for performance evaluation, and this dataset could be further served as a benchmark dataset in this area.


**Weaknesses:**

-	Reporting the stability test results through the line chart may be clearer. Besides, there is still some space to report more empirical evaluation results.

**Questions:**

Is the collected dataset publicly available?

**Limitations:**

None.

---

> ### Author Rebuttal · Authors · 2023-08-09
>
> Thank you for the constructive and insightful comments. Thank you for pointing out the strengths of our paper. Your concerns are addressed in detail below:
>
> **W1: Reporting the stability test results through the line chart may be clearer.**
>
> Thanks for the advice. We have conducted extra experiments at temperatures 0.4, 0.6, 0.8, and 1.0. The line charts in Figure (a) in the response PDF show that the difference between the highest Pass@1 (47.66%) at temperature = 0.4 and the lowest one (44.30%) at 0.0 is slight and the changes of the other metrics are also slight. These results suggest that SheetCopilot achieves stable performances even if the GPT-3.5 API temperature changes from 0.0 to 1.0.
>
> **Q1: Is the collected dataset publicly available?**
>
> Yes. Our dataset has been publicized on PapersWithCode anonymously. The reviewer can also check our dataset (including the task spreadsheets, task instructions, and ground truths) in the supplementary zip file.
>
> We thank the reviewer again for the thorough review and feedback.

---

> ### Comment · Reviewer_bHxV · 2023-08-16
>
> I've read the authors' response and decide to keep my score.

---

### Author Rebuttal · Authors · 2023-08-10

**Global Response**

_The authors would like to thank all reviewers for their appreciation and instructive suggestions!_

The authors are encouraged to hear that the reviews commented that

- the studied spreadsheet automation task is **valuable** (bHxV) and **challenging** (TF2z)
- the proposed SheetCopilot makes sense and is **novel** (bHxV), **inspiring** (XbAE), and **of big benefit** to spreadsheet users (EedW)
- our dataset curation process is **systematic** (TF2z), and our dataset is **a useful benchmark** for similar works (bHxV and HU2o)
- the ablation studies are **quite interesting** (XbAE) and **clearly show** our contributions (HU2o)
- our paper is **very well written** (HU2o) and the ideas are **neatly presented** (XbAE)
- no ethical issues appear in our work (Hbb1 and DGUT)

We carefully considered all concerns and comments provided by reviewers and addressed all of them appropriately. Our responses are summarized below:

**Q1 (XbAE and TF2z): Performances of smaller open-source LLMs.**

We tested smaller LLMs (e.g. llama13b, 7b) but found them insufficiently capable of generating complete solutions. Nevertheless, it is an intriguing research direction to develop small LLMs capable of solving software automation tasks.

**Q2 (TF2z and DGUT): Why GPT-3.5 is better than GPT-4.**

We investigated why GPT-3.5 is better than GPT-4 in detail (Please see lines 265 - 268 in the main paper; Supplementary Section F).

To clarify, GPT-4 does surpass GPT-3.5 in task success rate (Pass@1) as shown in Table 1. The lower Exec@1 of GPT-4 results from the experimental setting. To match GPT-3.5's 4096 token limit, we capped GPT-4's prompt plus response at 4096 tokens. Exceeding the token limit causes execution failure. This renders GPT-4 prone to incomplete solutions, while GPT-3.5's short yet incorrect outputs suffer less from the limit. Consequently, GPT-3.5 achieves a higher execution success rate (Exec@1) despite GPT-4's superior Pass@1.

In summary, GPT-4 outperforms GPT-3.5 in Pass@1 but the token limit renders GPT-4 less performant in Exec@1.

**Q3 (XbAE and TF2z): A more detailed discussion about the benchmark creation process and the nature of the dataset.**

Section A (Supplementary) details our benchmark creation process. Briefly, we collected ~27k SuperUser Q&A pairs, filtered irrelevant pairs by keywords, further cleaned them by prompting ChatGPT to label invalid pairs, and selected representatives through clustering, resulting in a clean Q&A pair dataset.

Figures A, B, and C (Supplementary) demonstrate our dataset's diverse complexity across six critical task categories.

We took deduplication measures to guarantee that our dataset is unseen by LLMs like GPT-3.5 (Section A.3 in the supplementary). The cosine similarity and ROUGE-L between our dataset and the SuperUser where our raw data comes from are < 0.16 and  < 0.35, respectively. This suggests that the SheetCopilot performances shown in Tables 1 and 2 are solid, without data leakage issues.

Overall, we believe our benchmark enables assessing LLM planning capabilities and will inspire further research at the intersection of language models and software agents.

**Q4 (TF2z and HU2o): About further investigation and failure analysis.**

We performed failure analysis for the tested LLMs by categorizing failure cases (Section F in the supplementary).

Key findings:

- GPT-3.5-Turbo struggles with formula prediction.
- GPT-4 often generates incomplete solutions due to token limits.
- Claude rarely repeats output but misapplies formulas/ranges.
- All models frequently neglect absolute references when auto-filling ranges, causing calculation errors.

The failure statistics also demonstrate interesting findings: GPT-3.5-Turbo and GPT-4 share similar failure patterns while Claude differs. This finding suggests that the two GPT models possess almost identical alignment in terms of spreadsheet manipulation and that a gap exists between the alignment of Claude and those of the GPT models.

We believe these results help researchers better understand the strengths and weaknesses of the leading LLMs.

**Q5 (TF2z and DGUT) While it is unlikely that one may see issues with spreadsheet data in discrimination or bias or fairness it would be good to address them.**

Thank you for bringing this to our attention. We understand the importance of fairness and unbiasedness in data. Rigorous steps were taken to uphold these principles:

Firstly, we removed any personally identifiable information from the collected spreadsheets to prevent discrimination. For example, fake names (e.g. numbers and letters) are used in the spreadsheets. Besides, the financial data in the spreadsheets are fabricated to guarantee that the sheets do not involve real individuals' or companies' financial information. These help to reduce bias, as it prevents LLMs from being influenced by personal characteristics.

Additionally, the authors thoroughly audited data, eliminating any offensive or biased content like racism or regional discrimination. For example, we removed tasks that process staff data according to country. One such task is “organizing them by name, country, and region within that country” which was labeled as “invalid” and discarded.

In summary, we implemented measures to address ethical concerns around bias and discrimination. We hope this reply addresses the reviewers’ concerns, and we are open to further suggestions to improve the ethical integrity of our work.

_Finally, we again show our greatest gratitude to all reviewers for their considerable and insightful comments._

---

### Decision · Program_Chairs · 2023-09-21

**Decision:**

Accept (poster)

**Comment:**

The submission presents a observe/propose/revise/act framework for using LLMs to interact with an existing spreadsheet application. The LLM is embedded in a simple state machine used to control the transition between different phases. The submission also describes a new dataset of spreadsheet-related tasks that is used for evaluation.

The importance of the problem was highlighted by reviewers, though some concerns about whether the implied latency of the presented system (requiring several LLM queries) makes it impractical. The paper is executed well, with reviewers highlighting the quality of writing, the ablation studies and the usefulness of the collected dataset.

Overall, reviews were positive (with the exception of TF2z, who did not engage with the rebuttal, but mainly raised weaknesses also covered by other reviewers) but not enthusiastic due to the somewhat incremental nature of the work.